# ADAM⁺: A STOCHASTIC METHOD WITH ADAPTIVE VARIANCE REDUCTION

## ABSTRACT

Adam is a widely used stochastic optimization method for deep learning applications. While practitioners prefer Adam because it requires less parameter tuning, its use is problematic from a theoretical point of view since it may not converge. Variants of Adam have been proposed with provable convergence guarantee, but they tend not be competitive with Adam on the practical performance. In this paper, we propose a new method named Adam⁺ (pronounced as Adam-plus). Adam⁺ retains some of the key components of Adam but it also has several noticeable differences: (i) it does not maintain the moving average of second moment estimate but instead computes the moving average of first moment estimate at extrapolated data points; (ii) its adaptive step size is formed not by dividing the square root of second moment estimate but instead by dividing the root of the norm of first moment estimate. As a result, Adam⁺ requires few parameter tuning, as Adam, but it enjoys a provable convergence guarantee. Our analysis further shows that Adam⁺ enjoys adaptive variance reduction, i.e., the variance of the stochastic gradient estimator reduces as the algorithm converges, hence enjoying an adaptive convergence. We also propose a more general variant of Adam⁺ with different adaptive step sizes and establish their fast convergence rate. Our empirical studies on various deep learning tasks, including image classification, language modeling, and automatic speech recognition, demonstrate that Adam⁺ significantly outperforms Adam and achieves comparable performance with best-tuned SGD and momentum SGD.

## 1 INTRODUCTION

Adaptive gradient methods (Duchi et al., 2011; McMahan & Streeter, 2010; Tieleman & Hinton, 2012; Kingma & Ba, 2014; Reddi et al., 2019) are one of the most important variants of Stochastic Gradient Descent (SGD) in modern machine learning applications. Contrary to SGD, adaptive gradient methods typically require little parameter tuning still retaining the computational efficiency of SGD. One of the most used adaptive methods is Adam (Kingma & Ba, 2014), which is considered by practitioners as the de-facto default optimizer for deep learning frameworks. Adam computes the update for every dimension of the model parameter through a moment estimation, i.e., the estimates of the first and second moments of the gradients. The estimates for first and second moments are updated using exponential moving averages with two different control parameters. These moving averages are the key difference between Adam and previous adaptive gradient methods, such as Adagrad (Duchi et al., 2011).

Although Adam exhibits great empirical performance, there still remain many mysteries about its convergence. First, it has been shown that Adam may not converge for some objective functions (Reddi et al., 2019; Chen et al., 2018b). Second, it is unclear what is the benefit that the moving average brings from theoretical point of view, especially its effect on the convergence rate. Third, it has been empirically observed that adaptive gradient methods can have worse generalization performance than its non-adaptive counterpart (e.g., SGD) on various deep learning tasks due to the coordinate-wise learning rates (Wilson et al., 2017).

The above issues motivate us to design a new algorithm which achieves the best of both worlds, i.e., provable convergence with benefits from the moving average and enjoying good generalization performance in deep learning. Specifically, we focus on the following optimization problem:

$$\min_{\mathbf{w} \in \mathbb{R}^d} F(\mathbf{w}),$$

Table 1: Summary of different algorithms with different assumptions and complexity results for finding an $\epsilon$-stationary point. "Individual Smooth" means assuming that $F(\mathbf{w}) = \mathbb{E}_{\xi \sim \mathcal{D}}[f(\mathbf{w}; \xi)]$ and that every component function $f(\mathbf{w}; \xi)$ is $L$-smooth. "Hessian Lipschitz" means that $\|\nabla^2 F(\mathbf{x}) - \nabla^2 F(\mathbf{y})\| \leq L_H \|\mathbf{x} - \mathbf{y}\|$ holds for $\mathbf{x}, \mathbf{y}$ and $L_H \geq 0$. "Type I" means that the complexity depends on $\mathbb{E}[\sum_{i=1}^{T} \|\mathbf{g}_{1:T,i}\|]$, where $\mathbf{g}_{1:T,i}$ stands for the $i$-th row of the matrix $[\mathbf{g}_1, \ldots, \mathbf{g}_T]$ with $\mathbf{g}_t$ being the stochastic gradient at $t$-th iteration and $T$ being the number of iterations. "Type II" means that complexity depends on $\mathbb{E}[\sum_{t=1}^{T} \|\mathbf{z}_t\|]$, where $\mathbf{z}_t$ is the variance-reduced gradient estimator at $t$-th iteration.

| Algorithm | Individual Smooth | Hessian Lipschitz | Worst-case Complexity better than $O(\epsilon^{-4})$? | Data-dependent Complexity |
|---|---|---|---|---|
| Generalized Adam (Chen et al., 2018b) PAdam (Chen et al., 2018a) Stagewise Adagrad (Chen et al., 2019) SGD with Adaptive Stepsize (Li & Orabona, 2019) Adagrad-Norm (Ward et al., 2019) | No | No | No | Type I |
| SPIDER (Fang et al., 2018) STORM (Cutkosky & Orabona, 2019) SNVRG (Zhou et al., 2018) Prox-SARAH (Pham et al., 2020) | Yes | No | Yes | N/A |
| SGD (Fang et al., 2019) Normalized momentum SGD (Cutkosky & Mehta, 2020) | No | Yes | Yes | N/A |
| Adam$^+$ (this work) | No | Yes | Yes | Type II |

where we only have access to stochastic gradients of $F$. Note that $F$ could possibly be nonconvex in $\mathbf{w}$. Due to the non-convexity, our goal is to design a stochastic first-order algorithm to find the $\epsilon$-stationary point, i.e., finding $\mathbf{w}$ such that $\mathbb{E}\left[\|\nabla F(\mathbf{w})\|\right] \leq \epsilon$, with low iteration complexity.

Our key contribution is the design and analysis of a new stochastic method named Adam$^+$. Adam$^+$ retains some of the key components of Adam but it also has several noticeable differences: (i) it does not maintain the moving average of second moment estimate but instead computes the moving average of first moment estimate at extrapolated points; (ii) its adaptive step size is formed not by dividing the square root of coordinate-wise second moment estimate but instead by dividing the root of the norm of first moment estimate. These features allow us to establish the adaptive convergence of Adam$^+$. Different from existing adaptive methods where the adaptive convergence depends on the growth rate of stochastic gradients (Duchi et al., 2011; McMahan & Streeter, 2010; Kingma & Ba, 2014; Luo et al., 2019; Reddi et al., 2019; Chen et al., 2018a;b; Ward et al., 2019; Li & Orabona, 2019; Chen et al., 2019), our adaptive convergence is due to the adaptive variance reduction property of our first order moment estimate. In existing literature, the variance reduction is usually achieved by large mini-batch (Goyal et al., 2017) or recursive variance reduction (Fang et al., 2018; Zhou et al., 2018; Pham et al., 2020; Cutkosky & Orabona, 2019). In contrast, we do not necessarily require large minibatch or computing stochastic gradients at two points per-iteration to achieve the variance reduction. In addition, we also establish a fast rate that matches the state-of-the-art complexity under the same conditions of a variant of Adam$^+$. Table 1 provides an overview of our results and a summary of existing results. We refer readers to Section F for a comprehensive survey of other related work. We further corroborate our theoretical results with an extensive empirical study on various deep learning tasks.

Our contributions are summarized below.

- We propose a new algorithm with adaptive step size, namely Adam$^+$, for general nonconvex optimization. We show that it enjoys a new type of data-dependent adaptive convergence that depends on the variance reduction property of first moment estimate. Notably, this data-dependent complexity does not require the presence of sparsity in stochastic gradients to guarantee fast convergence as in previous works (Duchi et al., 2011; Kingma & Ba, 2014; Reddi et al., 2019; Chen et al., 2019; 2018a). To the best of our knowledge, this is the first work establishing such new type of data-dependent complexity.

- We show that a general variant of our algorithm can achieve $O(\epsilon^{-3.5})$ worst-case complexity, which matches the state-of-the-art complexity guarantee under the Hessian Lipschitz assumption (Cutkosky & Mehta, 2020).

- We demonstrate the effectiveness of our algorithms on image classification, language modeling, and automatic speech recognition. Our empirical results show that our proposed algorithm consistently outperforms Adam on all tasks, and it achieves comparable performance with the best-tuned SGD and momentum SGD.

---

**Algorithm 1** Adam$^+$: Good default settings for the tested machine learning problems are $\alpha = 0.1, a = 1, \beta = 0.1, \epsilon_0 = 10^{-8}$.

---

1: **Require:** $\alpha, a \geq 1$: stepsize parameters
2: **Require:** $\beta \in (0, 1)$: Exponential decay rates for the moment estimate
3: **Require:** $g_t(\mathbf{w})$: unbiased stochastic gradient with parameters $\mathbf{w}$ at iteration $t$
4: **Require:** $\mathbf{w}_0$: Initial parameter vector
5: $\mathbf{z}_0 = g_0(\mathbf{w}_0)$
6: **for** $t = 0, \dots, T$ **do**
7:     Set $\eta_t = \frac{\alpha \beta^a}{\max(\|\mathbf{z}_t\|^{1/2}, \epsilon_0)}$
8:     $\mathbf{w}_{t+1} = \mathbf{w}_t - \eta_t \mathbf{z}_t$
9:     $\widehat{\mathbf{w}}_{t+1} = (1 - 1/\beta)\mathbf{w}_t + 1/\beta \cdot \mathbf{w}_{t+1}$
10:    $\mathbf{z}_{t+1} = (1 - \beta)\mathbf{z}_t + \beta g_{t+1}(\widehat{\mathbf{w}}_{t+1})$
11: **end for**

---

## 2 ALGORITHM AND THEORETICAL ANALYSIS

In this section, we introduce our algorithm Adam$^+$ (presented in Algorithm 1) and establish its convergence guarantees. Adam$^+$ resembles Adam in several aspects but also has noticeable differences. Similar to Adam, Adam$^+$ also maintains an exponential moving average of first moment (i.e., stochastic gradient), which is denoted by $\mathbf{z}_t$, and uses it for updating the solution in line 8. However, the difference is that the stochastic gradient is evaluated on an extrapolated data point $\widehat{\mathbf{w}}_{t+1}$, which is an extrapolation of two previous updates $\mathbf{w}_t$ and $\mathbf{w}_{t+1}$. Similar to Adam, Adam$^+$ also uses an adaptive step size that is proportional to $1/\|\mathbf{z}_t\|^{1/2}$. Nonetheless, the difference lies at its adaptive step size is directly computed from the square root of the norm of first moment estimate $\mathbf{z}_t$. In contrast, Adam uses an adaptive step size that is proportional to $1/\sqrt{\mathbf{v}_t}$, where $\mathbf{v}_t$ is an exponential moving average of second moment estimate. These two key components of Adam$^+$, i.e., extrapolation and adaptive step size from the root norm of the first moment estimate, make it enjoy two noticeable benefits: variance reduction of first moment estimate and adaptive convergence. We shall explain these two benefits later.

Before moving to the theoretical analysis, we would like to make some remarks. First, it is worth mentioning that the moving average estimate with extrapolation is inspired by the literature of stochastic compositional optimization (Wang et al., 2017). Wang et al. (2017) showed that the extrapolation helps balancing the noise in the gradients, reducing the bias in the estimates and giving a faster convergence rate. Here, our focus and analysis techniques are quite different. In fact, Wang et al. (2017) focuses on the compositional optimization while we consider a general nonconvex optimization setting. Moreover, the analysis in (Wang et al., 2017) mainly deals with the error of the gradient estimator caused by the compositional nature of the problem, while our analysis focuses on carefully designing adaptive normalization to obtain an adaptive and fast convergence rates. A similar extrapolation scheme has been also employed in the algorithm NIGT by Cutkosky & Mehta (2020). In later sections, we will also provide a more general variant of Adam$^+$ which subsumes NIGT as a special case.

Another important remark is that the update of Adam$^+$ is very different from the famous Nesterov's momentum method. In Nesterov's momentum method, the update of $\mathbf{w}_{t+1}$ uses the stochastic gradient at an extrapolated point $\widehat{\mathbf{w}}_{t+1} = \mathbf{w}_{t+1} + \gamma(\mathbf{w}_{t+1} - \mathbf{w}_t)$ with a momentum parameter $\gamma \in (0, 1)$. In contrast, in Adam$^+$ the update of $\mathbf{w}_{t+1}$ is using the moving average estimate at an extrapolated point $\widehat{\mathbf{w}}_{t+1} = \mathbf{w}_{t+1} + (1/\beta - 1)(\mathbf{w}_{t+1} - \mathbf{w}_t)$. Finally, Adam$^+$ does not employ coordinate-wise learning rates as in Adam, and hence it is expected to have better generalization performance according to Wilson et al. (2017).

### 2.1 ADAPTIVE VARIANCE REDUCTION AND ADAPTIVE CONVERGENCE

In this subsection, we analyze Adam$^+$ by showing its variance reduction property and adaptive convergence. To this end, we make the following assumptions.

**Assumption 1.** *There exists positive constants $L, \Delta, L_H, \sigma$ and an initial solution $\mathbf{w}_0$ such that*

    *(i) $F$ is $L$-smooth, i.e., $\|\nabla F(\mathbf{x}) - \nabla F(\mathbf{y})\| \leq L \|\mathbf{x} - \mathbf{y}\|, \ \forall \mathbf{x}, \mathbf{y} \in \mathbb{R}^d$.*

*(ii)* *For $\forall \mathbf{x} \in \mathbb{R}^d$, we have access to a first-order stochastic oracle at time $t$ $g_t(\mathbf{x})$ such that $\mathbb{E}\left[g_t(\mathbf{x})\right] = \nabla F(\mathbf{x})$, $\mathbb{E}\left\|g_t(\mathbf{x}) - \nabla F(\mathbf{x})\right\|^2 \leq \sigma^2$.*

*(iii)* *$\nabla F$ is a $L_H$-smooth mapping, i.e., $\|\nabla^2 F(\mathbf{x}) - \nabla^2 F(\mathbf{y})\| \leq L_H \|\mathbf{x} - \mathbf{y}\|, \forall \mathbf{x}, \mathbf{y} \in \mathbb{R}^d$.*

*(iv)* *$F(\mathbf{w}_0) - F_* \leq \Delta < \infty$, where $F_* = \inf_{\mathbf{w} \in \mathbb{R}^d} F(\mathbf{w})$.*

**Remark**: Assumption 1 (i) and (ii), (iv) are standard assumptions made in literature of stochastic non-convex optimization (Ghadimi & Lan, 2013). Assumption (iii) is the assumption that deviates from typical analysis of stochastic methods. We leverage this assumption to explore the benefit of moving average, extrapolation and adaptive normalization. It is also used in some previous works for establishing fast rate of stochastic first-order methods for nonconvex optimization (Fang et al., 2019; Cutkosky & Mehta, 2020) and this assumption is essential to get fast rate due to the hardness result in (Arjevani et al., 2019). It is also the key assumption for finding a local minimum in previous works (Carmon et al., 2018; Agarwal et al., 2017; Jin et al., 2017).

We might also assume that the stochastic gradient estimator in Algorithm 1 satisfies the following variance property.

**Assumption 2.** *Assume that $\mathbb{E}[\|g_0(\mathbf{w}_0) - \nabla F(\mathbf{w}_0)\|^2] \leq \sigma_0^2$ and $\mathbb{E}[\|g_t(\mathbf{w}_t) - \nabla F(\mathbf{w}_t)\|^2] \leq \sigma_m^2, t \geq 1$.*

**Remark:** When $g_0$ (resp. $g_t$) is implemented by a mini-batch stochastic gradient with mini-batch size $S$, then $\sigma_0^2$ (resp. $\sigma_m^2$) can be set as $\sigma^2/S$ by Assumption 1 (ii). We differentiate the initial variance and intermediate variance because they contribute differently to the convergence.

We first introduce a lemma to characterize the variance of the moving average gradient estimator $\mathbf{z}_t$.

**Lemma 1.** *Suppose Assumption 1 and Assumption 2 hold and $a \geq 1$. Then, there exists a sequence of random variables $\delta_t$ satisfying $\|\mathbf{z}_t - \nabla F(\mathbf{w}_t)\| \leq \delta_t$ for $\forall t \geq 0$,*

$$\mathbb{E}\left[\delta_{t+1}^2\right] \leq \left(1 - \frac{\beta}{2}\right) \mathbb{E}\left[\delta_t^2\right] + 2\beta^2 \sigma_m^2 + \mathbb{E}\left[\frac{CL_H^2 \|\mathbf{w}_{t+1} - \mathbf{w}_t\|^4}{\beta^3}\right]$$

$$\leq \left(1 - \frac{\beta}{2}\right) \mathbb{E}\left[\delta_t^2\right] + 2\beta^2 \sigma_m^2 + \mathbb{E}\left[CL_H^2 \alpha^4 \beta^{4a-3} \|\mathbf{z}_t\|^2\right],$$

*where $C = 1944$.*

**Remark**: Note that $\delta_t$ is an upper bound of $\|\mathbf{z}_t - \nabla F(\mathbf{w}_t)\|$, the above lemma can be used to illustrate the variance reduction effect for the gradient estimator $\mathbf{z}_t$. To this end, we can bound $\|\mathbf{z}_t\|^2 \leq 2\delta_t^2 + 2\|\nabla F(\mathbf{w}_t)\|^2$, then the term $CL^2 \alpha^4 \beta^{4a-3} \delta_t^2$ can be canceled with $-\beta/4 \delta_t^2$ with small enough $\alpha$. Hence, we have $\mathbb{E}\delta_{t+1}^2 \leq (1 - \beta/4)\mathbb{E}[\delta_t^2] + 2\beta^2 \sigma_m^2 + c\mathbb{E}[\|\nabla F(\mathbf{w}_t)\|^2]$ with a small constant $c$. As the algorithm converges with $\mathbb{E}[\|\nabla F(\mathbf{w}_t)\|^2]$ and $\beta$ decreases to zero, the variance of $\mathbf{z}_t$ will also decrease. Indeed, the above recursion of $\mathbf{z}_t$'s variance resembles that of the recursive variance reduced gradient estimators (e.g., SPIDER (Fang et al., 2018), STORM (Cutkosky & Orabona, 2019)). The benefit of using Adam$^+$ is that we do not need to compute stochastic gradient twice at each iteration.

We can now state our convergence rates for Algorithm 1.

**Theorem 1.** *Suppose Assumption 1 and Assumption 2 hold. Suppose $\|\nabla F(\mathbf{w})\| \leq G$ for any $\mathbf{w} \in \mathbb{R}^d$. By choosing the parameters such that $\alpha^4 \leq \frac{1}{36CL_H^2}$, $\alpha \leq \frac{1}{4L}$, $a = 1$ and $\epsilon_0 = \beta^a$, we have*

$$\frac{1}{T}\sum_{t=1}^{T} \mathbb{E}\left\|\nabla F(\mathbf{w}_t)\right\|^2 \leq \frac{G\mathbb{E}\left[\sum_{t=1}^{T}\|\mathbf{z}_t\|\right]}{T} + \frac{\Delta}{\alpha T} + \frac{18\sigma_0^2}{\beta T} + 30\beta\sigma_m^2 . \tag{1}$$

*In addition, suppose the initial batch size is $T_0$ and the intermediate batch size is $m$, and choose $\beta = T^{-b}$ with $0 \leq b \leq 1$, we have*

$$\frac{1}{T}\sum_{t=1}^{T} \mathbb{E}\left\|\nabla F(\mathbf{w}_t)\right\|^2 \leq \frac{\mathbb{E}\left[G\sum_{t=1}^{T}\|\mathbf{z}_t\|\right]}{T} + \frac{\Delta}{\alpha T} + \frac{18\sigma^2}{T^{1-b}T_0} + \frac{30\sigma^2}{mT^b} . \tag{2}$$

**Theorem 2.** *Suppose Assumption 1 and Assumption 2 hold. By choosing parameters such that* $640\alpha^3 L_H^{3/2} \le 1/120$, $a = 1$, $\epsilon_0 = 0$, $\beta = 1/T^s$ *with* $s = 2/3$ *then it takes* $T = O\left(\epsilon^{-4.5}\right)$ *number of iterations to ensure that*

$$\frac{1}{T}\sum_{t=1}^{T} \mathbb{E}\left[\|\nabla F(\mathbf{w}_t)\|^{3/2}\right] \le \epsilon^{3/2}, \quad \frac{1}{T}\mathbb{E}\left[\sum_{t=1}^{T} \delta_t^{3/2}\right] \le \epsilon^{3/2}\ .$$

**Remarks:**

- From Theorem 1, we can observe that the convergence rate of Adam$^+$ crucially depends on the growth rate of $\mathbb{E}\left[\sum_{t=1}^{T}\|\mathbf{z}_t\|\right]$, which gives a data-dependent adaptive complexity. If $\mathbb{E}\left[\sum_{t=1}^{T}\|\mathbf{z}_t\|\right] \le T^\alpha$ with $\alpha < 1$, then the algorithm converges. Smaller $\alpha$ implies faster convergence. Our goal is to ensure that $\frac{1}{T}\sum_{t=1}^{T}\mathbb{E}\|\nabla F(\mathbf{w}_t)\|^2 \le \epsilon^2$. Choosing $b = 1 - \alpha$, $m = O(1)$ and $T_0 = T^{1-\alpha} = O(\epsilon^{-2})$, and we end up with $T = O\left(\epsilon^{-\frac{2}{1-\alpha}}\right)$ complexity.

- Theorem 2 shows that in the ergodic sense, the Algorithm Adam$^+$ always converges, and the variance gets smaller when the number of iteration gets larger. Theorem 2 rules out the case that the magnitude of $\mathbf{z}_t$ converges to a constant and the bound (2) in Theorem 1 becomes vacuous.

- To compare with Adam-style algorithms (e.g., Adam, AdaGrad), these algorithms' convergence depend on the growth rate of stochastic gradient, i.e., $\sum_{i=1}^{d}\|\mathbf{g}_{1:T,i}\|/T$, where $\mathbf{g}_{1:T,i} = [g_{1,i}, \ldots, g_{T,i}]$ denotes the $i$-th coordinate of all historical stochastic gradients. Hence, the data determines the growth rate of stochastic gradient. If the stochastic gradients are not sparse, then its growth rate may not be slow and these Adam-style algorithms may suffer from slow convergence. In contrast, for Adam$^+$ the convergence can be accelerated by the variance reduction property. Note that we have $\mathbb{E}\left[\sum_{t=1}^{T}\|\mathbf{z}_t\|\right]/T \le \mathbb{E}\left[\sum_{t=1}^{T}(\delta_t + \|\nabla F(\mathbf{w}_t)\|)\right]/T$. Hence, Adam$^+$'s convergence depends on the variance reduction property of $\mathbf{z}_t$.

## 2.2 A General Variant of Adam$^+$: Fast Convergence with Large Mini-batch

Next, we introduce a more general variant of Adam$^+$ by making a simple change. In particular, we keep all steps the same as in Algorithm 1 except the adaptive step size is now set as $\eta_t = \frac{\alpha\beta^a}{\max(\|\mathbf{z}_t\|^p, \epsilon_0)}$, where $p \in [1/2, 1)$ is parameter. We refer to this general variant of Adam$^+$ as power normalized Adam$^+$ (Nadam$^+$). This generalization allows us to compare with some existing methods and to establish fast convergence rate. First, we notice that when setting $p = 1$ and $a = 5/4$ and $\beta = 1/T^{4/7}$, Nadam$^+$ is almost the same as the stochastic method NIGT (Cutkosky & Mehta, 2020) with only some minor differences. However, we observed that normalizing by $\|\mathbf{z}_t\|$ leads to slow convergence in practice, so we are instead interested in $p < 1$. Below, we will show that NAdam$^+$ with $p < 1$ can achieve a fast rate of $1/\epsilon^{3.5}$, which is the same as NIGT.

**Theorem 3.** *Under the same assumption as in Theorem 1, further assume* $\sigma_0^2 = \sigma^2/T_0$ *and* $\sigma_m^2 = \sigma^2/m$. *By using the step size* $\eta_t = \frac{\alpha\beta^{4/3}}{\max(\|\mathbf{z}_t\|^{2/3}, \epsilon_0)}$ *in Algorithm 1 with* $CL^2\alpha^4 \le 1/14$, $\epsilon_0 = 2\beta^{4/3}$, *in order to have* $\mathbb{E}\left[\|\nabla F(\mathbf{w}_\tau)\|\right] \le \epsilon$ *for a randomly selected solution* $\mathbf{w}_\tau$ *from* $\{\mathbf{w}_1, \ldots, \mathbf{w}_T\}$, *it suffice to set* $\beta = O(\epsilon^{1/2})$, $T = O(\epsilon^{-2})$, *the initial batch size* $T_0 = 1/\beta = O(\epsilon^{-1/2})$, *the intermediate batch size as* $m = 1/\beta^3 = O(\epsilon^{-3/2})$, *which ends up with the total complexity* $O(\epsilon^{-3.5})$.

**Remark:** Note that the above theorem establishes the fast convergence rate for Nadam$^+$ with $p = 2/3$. Indeed, we can also establish a fast rate of Adam$^+$ (where $p = 1/2$) in the order of $O(1/\epsilon^{3.625})$ with details provided in the Appendix E.

## 3 Experiments

In this section, we conduct empirical studies to verify the effectiveness of the proposed algorithm on three different tasks: image classification on CIFAR10 and CIFAR100 dataset (Krizhevsky et al.,

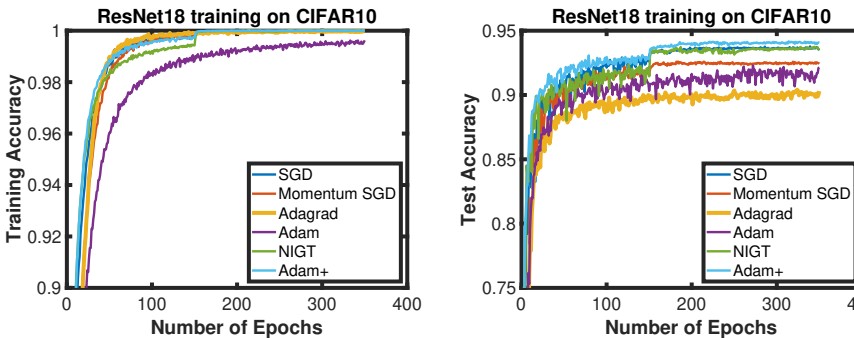

Figure 1: Comparison of optimization methods for ResNet18 Training on CIFAR10.

2009), language modeling on Wiki-Text2 dataset (Merity, 2016) and automatic speech recognition on SWB-300 dataset (Saon et al., 2017). We choose tasks from different domains to demonstrate the applicability for the real-world deep learning tasks in a broad sense. The detailed description is presented in Table 2. We compare our algorithm Adam$^+$ with SGD, momentum SGD, Adagrad, NIGT and Adam. We choose the same random initialization for each algorithm, and run a fixed number of epochs for every task. For Adam we choose the default setting $\beta_1 = 0.9$ and $\beta_2 = 0.999$ as in the original Adam paper.

Table 2: Summary of setups in the experiments.

| Domain | Task | Architecture | Dataset |
|---|---|---|---|
| Computer Vision | Image Classification | ResNet18 | CIFAR10 |
| Computer Vision | Image Classification | VGG19 | CIFAR100 |
| Natural Language Processing | Language Modeling | Two-layer LSTM | Wiki-Text2 |
| Automatic Speech Recognition | Speech Recognition | Six-layer BiLSTM | SWB-300 |

### 3.1 IMAGE CLASSIFICATION

**CIFAR10 and CIFAR100** In the first experiment, we consider training ResNet18 (He et al., 2016) and VGG19 (Simonyan & Zisserman, 2014) to do image classification task on CIFAR10 and CI-FAR100 dataset respectively. For every optimizer, we use batch size 128 and run 350 epochs. For SGD and momentum SGD, we set the initial learning rate to be $0.1$ for the first 150 epochs, and the learning rate is decreased by a factor of 10 for every 100 epochs. For Adagrad and Adam, the initial learning rate is tuned from $\{0.1, 0.01, 0.001\}$ and we choose the one with the best performance. The best initial learning rates for Adagrad and Adam are $0.01$ and $0.001$ respectively. For NIGT, we tune the their momentum parameter from $\{0.01, 0.1, 0.9\}$ (the best momentum parameter we found is $0.9$) and the learning rate is chosen the same as in SGD. For Adam$^+$, the learning rate is set according to Algorithm 1, in which we choose $\beta = 0.1$ and the value of $\alpha$ is the same as the learning rate used in SGD. We report training and test accuracy versus the number of epochs in Figure 1 for CIFAR10 and Figure 2 for CIFAR100. We observe that our algorithm consistently outperforms all other algorithms on both CIFAR10 and CIFAR100, in terms of both training and testing accuracy. Notably, we have some interesting observations for the training of VGG19 on CIFAR100. First, both Adam$^+$ and NIGT significantly outperform SGD, momentum SGD, Adagrad and Adam. Second, Adam$^+$ achieves almost the same final accuracy as NIGT, and Adam$^+$ converges much faster in the early stage of the training.

### 3.2 LANGUAGE MODELING

**Wiki-text2** In the second experiment, we consider the language modeling task on WikiText-2 dataset. We use a 2-layer LSTM (Hochreiter & Schmidhuber, 1997). The size of word embeddings is 650 and the number of hidden units per layer is 650. We run every algorithm for 40 epochs, with batch size 20 and dropout ratio 0.5. For SGD and momentum SGD, we tune the initial learning rate from $\{0.1, 0.2, 0.5, 5, 10, 20\}$ and decrease the learning rate by factor of 4 when the validation error

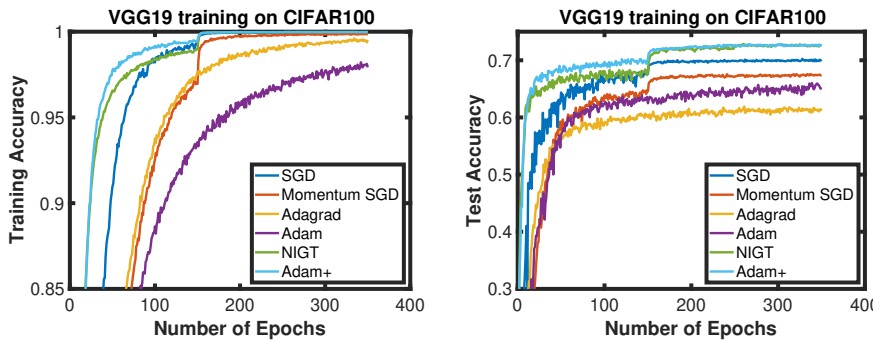

Figure 2: Comparison of optimization methods for VGG19 training on CIFAR100.

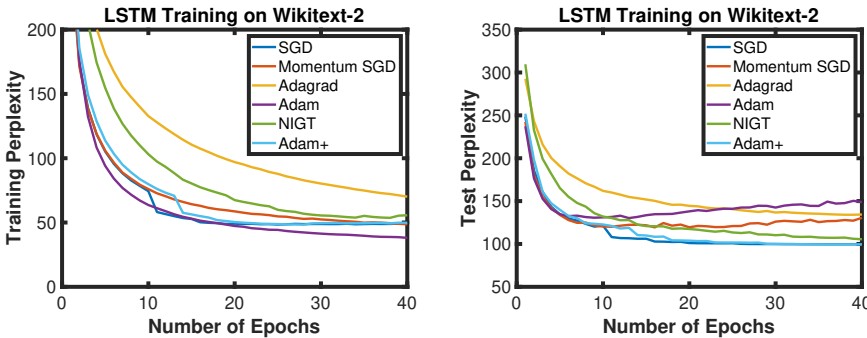

Figure 3: Comparison of optimization methods for two-layers LSTM training on WikiText-2.

saturates. For Adagrad and Adam, we tune the initial learning rate from $\{0.001, 0.01, 0.1, 1.0\}$. We report the best performance for these methods across the range of learning rate. The best initial learning rates for Adagrad and Adam are 0.01 and 0.001 respectively. For NIGT, we tune the initial value of learning rate from the same range as in SGD, and tune the momentum parameter $\beta$ from $\{0.01, 0.1, 0.9\}$, and the best parameter choice is $\beta = 0.9$. The learning rate and $\beta$ are both decreased by a factor of $4$ when the validation error saturates. For Adam$^+$, we follow the same tuning strategy as NIGT.

We report both training and test perplexity versus the number of epochs in Figure 3. From the Figure, we have the following observations: First, in terms of training perplexity, our algorithm achieves comparable performance with SGD and momentum SGD and outperforms Adagrad and NIGT, and it is worse than Adam. Second, in terms of test perplexity, our algorithm outperforms Adam, Adagrad, NIGT and momentum SGD, and it is comparable to SGD. An interesting observation is that Adam does not generalize well even if it has fast convergence in terms of training error, which is consistent with the observations in (Wilson et al., 2017).

### 3.3 AUTOMATIC SPEECH RECOGNITION

**SWB-300** In the third experiment, we consider the automatic speech recognition task on SWB-300 dataset (Saon et al., 2017). SWB-300 contains roughly 300 hours of training data of over 4 million samples (30GB) and roughly 6 hours of held-out data of over 0.08 million samples (0.6GB). Each training sample is a fusion of FMLLR (40-dim), i-Vector (100-dim), and logmel with its delta and double delta. The acoustic model is a long short-term memory (LSTM) model with 6 bi-directional layers. Each layer contains 1,024 cells (512 cells in each direction). On top of the LSTM layers, there is a linear projection layer with 256 hidden units, followed by a softmax output layer with 32,000 (i.e., 32,000 classes) units corresponding to context-dependent HMM states. The LSTM is unrolled with 21 frames and trained with non-overlapping feature sub-sequences of that length. This model contains over 43 million parameters and is about 165MB large. The training takes about 20 hours on 1 V100 GPU. To compare, we adopt the well-tuned Momentum SGD strategy as described in (Zhang et al., 2019) for this task as the baseline: batch size is 256, learning rate is

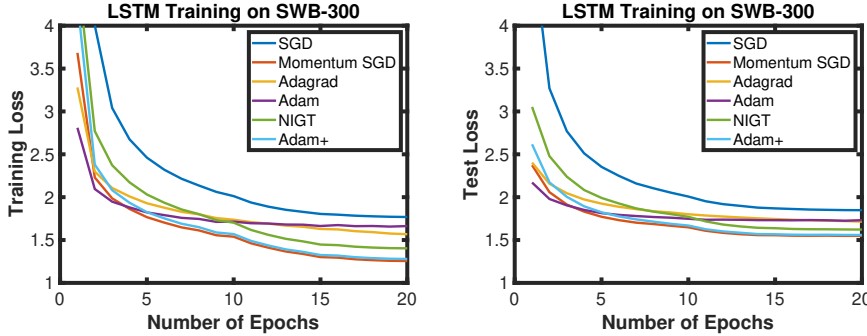

Figure 4: Comparison of optimization methods for six-layers LSTM training on SWB-300.

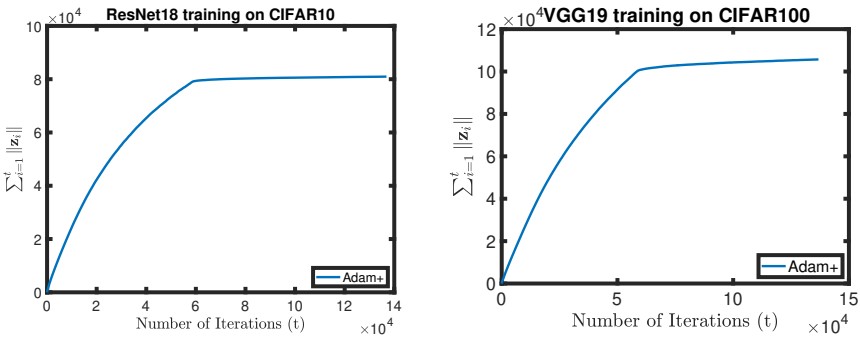

Figure 5: The growth of quantity $\sum_{i=1}^{t} \|\mathbf{z}_i\|$ in Adam$^+$

0.1 for the first 10 epochs and then annealed by $\sqrt{0.5}$ for another 10 epochs, with momentum 0.9. We grid search the learning rate of Adam and Adagrad from $\{0.1, 0.01, 0.001\}$, and report the best configuration we have found (Adam with learning rate 0.001 and Adagrad with learning rate 0.01). For NIGT, we also follow the same learning rate setup (including annealing) as in Momentum SGD baseline. In addition, we fine tuned $\beta$ in NIGT by exploring $\beta$ in $\{0.01, 0.1, 0.9\}$ and reported the best configuration ($\beta$=0.9). For Adam$^+$, we follow the same learning rate and annealing strategy as in the Momentum SGD and tuned $\beta$ in the same way as in NGIT, reporting the best configuration ($\beta$=0.01). From Figure 4, Adam$^+$ achieves the indistinguishable training loss and held-out loss w.r.t. well-tuned Momentum SGD baseline and significantly outperforms the other optimizers.

### 3.4 GROWTH RATE OF $\sum_{i=1}^{t} \|\mathbf{z}_i\|$

In this subsection, we consider the growth rate of $\sum_{i=1}^{t} \|\mathbf{z}_i\|$, since they crucially affect the convergence rate as shown in Theorem 1. We report the results of both ResNet18 training on CIFAR10 dataset and VGG19 training on CIFAR100 dataset. From Figure 5, we can observe that it quickly reaches a plateau and then grows at a very slow rate with respect to the number of iterations. This phenomenon verifies the variance reduction effect and also explains the reason why Adam$^+$ enjoys a fast convergence speed in practice.

## 4 CONCLUSION

In this paper, we design a new algorithm named Adam$^+$ to train deep neural networks efficiently. Different from Adam, Adam$^+$ updates the solution using moving average of stochastic gradients calculated at the extrapolated points and adaptive normalization on only first-order statistics of stochastic gradients. We establish data-dependent adaptive complexity results for Adam$^+$ from the perspective of adaptive variance reduction, and also show that a variant of Adam$^+$ achieves state-of-the-art complexity. Extensive empirical studies on several tasks verify the effectiveness of the proposed algorithm. We also empirically show that the slow growth rate of the new gradient estimator, providing the reason why Adam$^+$ enjoys fast convergence in practice.

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

# A   PROOF OF LEMMA 1

*Proof.* The proof is similar to that of Lemma 12 in (Wang et al., 2017). Define

$$\zeta_k^{(t)} = \begin{cases} \beta(1-\beta)^{t-k} & \text{if } t \geq k > 0 \\ (1-\beta)^{t-k} & \text{if } t \geq k = 0 \end{cases} \tag{3}$$

By the definition of $\zeta_t^{(k)}$ and the update of Algorithm 1, we have

$$\zeta_k^{(t+1)} = (1-\beta)\zeta_k^{(t)}, \quad \sum_{k=0}^t \zeta_k^{(t)} = 1, \quad \mathbf{w}_t = \sum_{k=0}^t \zeta_k^{(t)} \widehat{\mathbf{w}}_{t+1}, \quad \mathbf{z}_{t+1} = \sum_{k=0}^t \zeta_k^{(t)} \nabla f(\widehat{\mathbf{w}}_{t+1}; \xi_{t+1}).$$

Define $m_{t+1} = \sum_{k=0}^t \zeta_k^{(t)} \|\mathbf{w}_{t+1} - \widehat{\mathbf{w}}_{k+1}\|^2$, $n_{t+1} = \sum_{k=0}^t \zeta_k^{(t)} [\nabla f(\widehat{\mathbf{w}}_{k+1}; \xi_{k+1}) - F(\widehat{\mathbf{w}}_{k+1})]$, where $\nabla f(\widehat{\mathbf{w}}_{k+1}; \xi_{k+1})$ is an unbiased stochastic first-order oracle for $F(\widehat{\mathbf{w}}_{k+1})$ with bounded variance $\sigma_m^2$. Note that $\nabla F$ is a $L_H$-smooth mapping (according to Assumption 1 (iii)), then by Lemma 10 of (Wang et al., 2017), we have

$$\|\mathbf{z}_t - \nabla F(\mathbf{w}_t)\|^2 \leq (L_H m_t + \|n_t\|)^2 \leq 2L_H^2 m_t^2 + 2\|n_t\|^2.$$

Define $q_{t+1} = \sum_{k=0}^t \zeta_k^{(t)} \|\mathbf{w}_{t+1} - \widehat{\mathbf{w}}_{k+1}\|$. According to Lemma 11 (a) and (b) of (Wang et al., 2017), we have

$$m_{t+1} + 4q_{t+1}^2 \leq \left(1 - \frac{\beta}{2}\right)(m_t + 4q_t^2) + \frac{18}{\beta}\|\mathbf{w}_{t+1} - \mathbf{w}_t\|^2.$$

Taking squares on both sides of the inequality and using the fact that $(a+b)^2 \leq (1+\frac{\beta}{2})a^2 + (1+\frac{2}{\beta})b^2$ for $\beta > 0$, we have

$$\begin{aligned} \left(m_{t+1} + 4q_{t+1}^2\right)^2 &\leq \left(1 + \frac{\beta}{2}\right)\left(1 - \frac{\beta}{2}\right)^2 (m_t + 4q_t^2)^2 + \left(1 + \frac{2}{\beta}\right)\frac{324}{\beta^2}\|\mathbf{w}_{t+1} - \mathbf{w}_t\|^4 \\ &\leq \left(1 - \frac{\beta}{2}\right)(m_t + 4q_t^2)^2 + \frac{972}{\beta^3}\|\mathbf{w}_{t+1} - \mathbf{w}_t\|^4, \end{aligned} \tag{4}$$

where the last inequality holds since $1/\beta \geq 1$.

Define $\delta_t^2 = 2L_H^2(m_t + 4q_t^2)^2 + 2\|n_t\|^2$, then we have $\|\mathbf{z}_t - \nabla F(\mathbf{w}_t)\|^2 \leq \delta_t^2$ for all $t$. Denote $\mathcal{F}_{t+1}$ by the $\sigma$-algebra generated by $\xi_1, \ldots, \xi_{t+1}$. Taking the summation of (4) and according to the bound of $n_t$ derived in Lemma 11 (c) of (Wang et al., 2017), we have

$$\mathbb{E}\left[\delta_{t+1}^2 | \mathcal{F}_{t+1}\right] \leq \left(1 - \frac{\beta}{2}\right)\delta_t^2 + 2\beta^2\sigma_m^2 + \frac{1944 L_H^2 \|\mathbf{w}_{t+1} - \mathbf{w}_t\|^4}{\beta^3},$$

Taking expectation on both sides yields

$$\mathbb{E}\left[\delta_{t+1}^2\right] \leq \left(1 - \frac{\beta}{2}\right)\mathbb{E}\left[\delta_t^2\right] + 2\beta^2\sigma_m^2 + \mathbb{E}\left[\frac{1944 L_H^2 \|\mathbf{w}_{t+1} - \mathbf{w}_t\|^4}{\beta^3}\right].$$

Note that $\eta_t = \frac{\alpha\beta^a}{\max(\|\mathbf{z}_t\|^{1/2}, \epsilon_0)}$, we have

$$\mathbb{E}\left[\delta_{t+1}^2\right] \leq \left(1 - \frac{\beta}{2}\right)\mathbb{E}\left[\delta_t^2\right] + 2\beta^2\sigma_m^2 + \mathbb{E}\left[CL^2\alpha^4\beta^{4a-3}\|\mathbf{z}_t\|^2\right]. \qquad \square$$

# B   PROOF OF THEOREM 1

*Proof.* By Lemma 1 and the update rule of Algorithm 1, we have

$$\begin{aligned} \mathbb{E}\left[\delta_{t+1}^2\right] &\leq \left(1 - \frac{\beta}{2}\right)\mathbb{E}\left[\delta_t^2\right] + 2\beta^2\sigma_m^2 + \mathbb{E}\left[\frac{CL_H^2\alpha^4\beta^{4a}\|\mathbf{z}_t\|^4}{\beta^3(\max(\|\mathbf{z}_t\|^{1/2}, \epsilon_0))^4}\right] \\ &\leq \left(1 - \frac{\beta}{2}\right)\mathbb{E}\left[\delta_t^2\right] + 2\beta^2\sigma_m^2 + \mathbb{E}\left[\frac{2CL_H^2\alpha^4\beta^4(\|\delta_t\|^2 + \|\nabla F(\mathbf{w}_t)\|^2)}{\beta^3}\right], \end{aligned} \tag{5}$$

where the second inequality holds since $(\max(\|\mathbf{z}_t\|^{1/2}, \epsilon_0))^4 \geq \|\mathbf{z}_t\|^2$ and $\|\mathbf{z}_t\|^2 \leq 2\|\delta_t\|^2 + 2\|\nabla F(\mathbf{w}_t)\|^2$.

Note that $2CL_H^2\alpha^4 \leq 1/18$. Plugging it in (5), we have

$$\frac{8\beta}{18}\mathbb{E}\left[\delta_t^2\right] \leq \mathbb{E}\left[\delta_t^2 - \delta_{t+1}^2\right] + 2\beta^2\sigma_m^2 + \mathbb{E}\left[\frac{\beta}{18}\|\nabla F(\mathbf{w}_t)\|^2\right]. \tag{6}$$

Summing over $t = 1, \ldots, T$ on both sides of (6) and with some simple algebra, we have

$$\sum_{t=1}^T \mathbb{E}\left[\delta_t^2\right] \leq \sum_{t=1}^T \mathbb{E}\left[\frac{3\left(\delta_t^2 - \delta_{t+1}^2\right)}{\beta}\right] + \sum_{t=1}^T 5\beta\sigma_m^2 + \sum_{t=1}^T \mathbb{E}\left[\frac{1}{8}\|\nabla F(\mathbf{w}_t)\|^2\right]. \tag{7}$$

By Assumption 1 (i) and by the property of $L$-smooth function, we know that

$$F(\mathbf{w}_{t+1}) \leq F(\mathbf{w}_t) + \nabla^\top F(\mathbf{w}_t)(\mathbf{w}_{t+1} - \mathbf{w}_t) + \frac{L}{2}\|\mathbf{w}_{t+1} - \mathbf{w}_t\|^2$$

$$= F(\mathbf{w}_t) - \eta_t\nabla^\top F(\mathbf{w}_t)\mathbf{z}_t + \frac{\eta_t^2 L}{2}\|\mathbf{z}_t\|^2$$

$$\leq F(\mathbf{w}_t) - \eta_t\nabla^\top F(\mathbf{w}_t)(\mathbf{z}_t - \nabla F(\mathbf{w}_t) + \nabla F(\mathbf{w}_t)) + \eta_t^2 L\left(\|\mathbf{z}_t - \nabla F(\mathbf{w}_t)\|^2 + \|\nabla F(\mathbf{w}_t)\|^2\right)$$

$$= F(\mathbf{w}_t) - (\eta_t - \eta_t^2 L)\|\nabla F(\mathbf{w}_t)\|^2 - \eta_t\nabla^\top F(\mathbf{w}_t)(\mathbf{z}_t - \nabla F(\mathbf{w}_t)) + \eta_t^2 L\|\mathbf{z}_t - F(\mathbf{w}_t)\|^2$$

$$\leq F(\mathbf{w}_t) - \left(\frac{\eta_t}{2} - \eta_t^2 L\right)\|\nabla F(\mathbf{w}_t)\|^2 + \left(\frac{\eta_t}{2} + \eta_t^2 L\right)\|\mathbf{z}_t - F(\mathbf{w}_t)\|^2.$$

Noting that $\eta_t = \frac{\alpha\beta^a}{\max\left(\|\mathbf{z}_t\|^{1/2}, \epsilon_0\right)}$, $\alpha \leq 1/4L$ and $\epsilon_0 = \beta^a$, we know that $\eta_t L \leq 1/4$. Hence we have

$$\|\nabla F(\mathbf{w}_t)\|^2 \leq \frac{4(F(\mathbf{w}_t) - F(\mathbf{w}_{t+1}))}{\eta_t} + 3\|\mathbf{z}_t - \nabla F(\mathbf{w}_t)\|^2.$$

Taking summation over $t = 1, \ldots, T$ and taking expectation yield

$$\sum_{t=1}^T \mathbb{E}\|\nabla F(\mathbf{w}_t)\|^2 \leq \mathbb{E}\left[\sum_{t=1}^T \frac{4(F(\mathbf{w}_t) - F(\mathbf{w}_{t+1}))}{\eta_t}\right] + 3\sum_{t=1}^T \mathbb{E}\|\mathbf{z}_t - \nabla F(\mathbf{w}_t)\|^2$$

$$\leq \mathbb{E}\left[\sum_{t=1}^T \frac{4(F(\mathbf{w}_t) - F(\mathbf{w}_{t+1}))}{\eta_t}\right] + 3\sum_{t=1}^T \mathbb{E}\left[\delta_t^2\right]. \tag{8}$$

Combining (7) and (8) yields

$$\sum_{t=1}^T \mathbb{E}\|\nabla F(\mathbf{w}_t)\|^2 \leq \mathbb{E}\left[\sum_{t=1}^T \frac{4(F(\mathbf{w}_t) - F(\mathbf{w}_{t+1}))}{\eta_t}\right] + \sum_{t=1}^T \mathbb{E}\left[\frac{9\left(\delta_t^2 - \delta_{t+1}^2\right)}{\beta}\right]$$

$$+ \sum_{t=1}^T 15\beta\sigma_m^2 + \sum_{t=1}^T \mathbb{E}\left[\frac{3}{8}\|\nabla F(\mathbf{w}_t)\|^2\right].$$

By some simple algebra, we have

$$\sum_{t=1}^T \mathbb{E}\|\nabla F(\mathbf{w}_t)\|^2 \leq \mathbb{E}\left[\sum_{t=1}^T \frac{8(F(\mathbf{w}_t) - F(\mathbf{w}_{t+1}))}{\eta_t}\right] + \sum_{t=1}^T \mathbb{E}\left[\frac{18\left(\delta_t^2 - \delta_{t+1}^2\right)}{\beta}\right] + \sum_{t=1}^T 30\beta\sigma_m^2.$$

Then we have

$$\frac{1}{T}\sum_{t=1}^T \mathbb{E}\|\nabla F(\mathbf{w}_t)\|^2 \leq \mathbb{E}\left[\sum_{t=1}^T \frac{8\max\left(\|\mathbf{z}_t\|^{1/2}, \epsilon_0\right)(F(\mathbf{w}_t) - F(\mathbf{w}_{t+1}))}{\alpha\beta^a T}\right] + \frac{18\sigma_0^2}{\beta T} + 30\beta\sigma_m^2. \tag{9}$$

Noting that $|F(\mathbf{w}_t) - F(\mathbf{w}_{t+1})| \leq G\eta_t\|\mathbf{z}_t\|$, we have

$$\frac{1}{T}\sum_{t=1}^T \mathbb{E}\|\nabla F(\mathbf{w}_t)\|^2 \leq \frac{8G\mathbb{E}\left[\sum_{t=1}^T \|\mathbf{z}_t\|\right]}{T} + \frac{\Delta}{\alpha T} + \frac{18\sigma_0^2}{\beta T} + 30\beta\sigma_m^2. \tag{10}$$

$\square$

## C  PROOF OF THEOREM 2

Before introducing the proof, we first introduce several lemmas which are useful for our analysis.

**Lemma 2.** *Adam$^+$ with $\eta_t = \frac{\alpha\beta}{\max(\|\mathbf{z}_t\|^{1/2}, \epsilon_0)}$ and $\epsilon_0 = 0$ satisfies*

$$F(\mathbf{w}_{t+1}) - F(\mathbf{w}_t) \leq \alpha\beta\left(-\frac{\|\nabla F(\mathbf{w}_t)\|^{3/2}}{6} + 9\|\mathbf{z}_t - \nabla F(\mathbf{w}_t)\|^{3/2}\right) + \frac{64\alpha^4\beta^4 L^3}{3}.$$

*Proof.* By the $L$-smoothness and the update of the algorithm, we have

$$\begin{aligned}
F(\mathbf{w}_{t+1}) - F(\mathbf{w}_t) &\leq \nabla^\top F(\mathbf{w}_t)(\mathbf{w}_{t+1} - \mathbf{w}_t) + \frac{L\|\mathbf{w}_{t+1} - \mathbf{w}_t\|^2}{2} \\
&\leq -\alpha\beta \cdot \frac{\langle \nabla F(\mathbf{w}_t), \mathbf{z}_t \rangle}{\max\left(\|\mathbf{z}_t\|^{1/2}, \epsilon_0\right)} + \frac{\alpha^2\beta^2 L\|\mathbf{z}_t\|^2}{\left(\max\left(\|\mathbf{z}_t\|^{1/2}, \epsilon_0\right)\right)^2}.
\end{aligned} \tag{11}$$

Define $\Delta_t = \mathbf{z}_t - \nabla F(\mathbf{w}_t)$. If $\|\nabla F(\mathbf{w}_t)\| \geq 2\|\Delta_t\|$, we have

$$\begin{aligned}
-\frac{\langle \mathbf{z}_t, \nabla F(\mathbf{w}_t)\rangle}{\max\left(\|\mathbf{z}_t\|^{1/2}, \epsilon_0\right)} &= -\frac{\|\nabla F(\mathbf{w}_t)\|^2 + \langle \Delta_t, \nabla F(\mathbf{w}_t)\rangle}{\max\left(\|\nabla F(\mathbf{w}_t) + \Delta_t\|^{1/2}, \epsilon_0\right)} \\
&\leq -\frac{\|\nabla F(\mathbf{w}_t)\|^2}{2\|\nabla F(\mathbf{w}_t) + \Delta_t\|^{1/2}} \leq -\frac{\|\nabla F(\mathbf{w}_t)\|^{3/2}}{3} \\
&\leq -\frac{\|\nabla F(\mathbf{w}_t)\|^{3/2}}{3} + 8\|\Delta_t\|^{3/2}.
\end{aligned} \tag{12}$$

If $\|\nabla F(\mathbf{w}_t)\| \leq 2\|\Delta_t\|$, we have

$$\begin{aligned}
-\frac{\langle \mathbf{z}_t, \nabla F(\mathbf{w}_t)\rangle}{\max\left(\|\mathbf{z}_t\|^{1/2}, \epsilon_0\right)} &= -\frac{\|\nabla F(\mathbf{w}_t)\|^2 + \langle \Delta_t, \nabla F(\mathbf{w}_t)\rangle}{\max\left(\|\nabla F(\mathbf{w}_t) + \Delta_t\|^{1/2}, \epsilon_0\right)} \\
&\leq \frac{6\|\Delta_t\|^2}{\|\Delta_t\|^{1/2}} = 6\|\Delta_t\|^{3/2} \leq -\frac{\|\nabla F(\mathbf{w}_t)\|^{3/2}}{3} + 8\|\Delta_t\|^{3/2}.
\end{aligned} \tag{13}$$

By (12) and (13), we have

$$-\frac{\langle \mathbf{z}_t, \nabla F(\mathbf{w}_t)\rangle}{\max\left(\|\mathbf{z}_t\|^{1/2}, \epsilon_0\right)} \leq -\frac{\|\nabla F(\mathbf{w}_t)\|^{3/2}}{3} + 8\|\Delta_t\|^{3/2}. \tag{14}$$

By (11) and (14), we have

$$\begin{aligned}
F(\mathbf{w}_{t+1}) - F(\mathbf{w}_t) &\leq \alpha\beta\left(\frac{\|\nabla F(\mathbf{w}_t)\|^{3/2}}{3} + 8\|\Delta_t\|^{3/2}\right) + \alpha^2\beta^2 L\|\mathbf{z}_t\| \\
&= \alpha\beta\left(-\frac{\|\nabla F(\mathbf{w}_t)\|^{3/2}}{3} + 8\|\Delta_t\|^{3/2}\right) + \alpha^2\beta^2 L \min_{x>0}\left(\frac{2\|\mathbf{z}_t\|^{3/2}}{3x} + \frac{x^2}{3}\right) \\
&\leq \alpha\beta\left(-\frac{\|\nabla F(\mathbf{w}_t)\|^{3/2}}{3} + 8\|\Delta_t\|^{3/2}\right) + \alpha^2\beta^2 L\left(\frac{2\|\mathbf{z}_t\|^{3/2}}{3(8\alpha\beta L)} + \frac{64\alpha^2\beta^2 L^2}{3}\right) \\
&\leq \alpha\beta\left(-\frac{\|\nabla F(\mathbf{w}_t)\|^{3/2}}{6} + 9\|\Delta_t\|^{3/2}\right) + \frac{64\alpha^4\beta^4 L^3}{3},
\end{aligned}$$

where the last inequality holds because $\|\mathbf{z}_t\|^{3/2} \leq 2\|\nabla F(\mathbf{w}_t)\|^{3/2} + 2\|\Delta_t\|^{3/2}$.  □

**Lemma 3.** *For Adam$^+$ with $\eta_t = \frac{\alpha\beta}{\max(\|\mathbf{z}_t\|^{1/2}, \epsilon_0)}$, there exist random variables $\delta_t$ such that*

$$\mathbb{E}\left[\delta_{t+1}^{3/2}\right] \leq \left(1 - \frac{\beta}{2}\right)\mathbb{E}\left[\delta_t^{3/2}\right] + 2\beta^{3/2}\sigma^{3/2} + \mathbb{E}\left[\frac{320 L_H^{3/2}\|\mathbf{w}_{t+1} - \mathbf{w}_t\|^3}{\beta^2}\right].$$

*Proof.* The proof shares the similar spirit of Lemma 12 in (Wang et al., 2017), but we adapt the proof for our purpose. Define

$$\zeta_k^{(t)} = \begin{cases} \beta(1-\beta)^{t-k} & \text{if } t \geq k > 0 \\ (1-\beta)^{t-k} & \text{if } t \geq k = 0 \end{cases} \tag{15}$$

By the definition of $\zeta_t^{(k)}$ and the update of Algorithm 1, we have

$$\zeta_k^{(t+1)} = (1-\beta)\zeta_k^{(t)}, \quad \sum_{k=0}^{t} \zeta_k^{(t)} = 1, \quad \mathbf{w}_t = \sum_{k=0}^{t} \zeta_k^{(t)} \widehat{\mathbf{w}}_{k+1}, \quad \mathbf{z}_{t+1} = \sum_{k=0}^{t} \zeta_k^{(t)} \nabla f(\widehat{\mathbf{w}}_{k+1}; \xi_{k+1}).$$

Define $m_{t+1} = \sum_{k=0}^{t} \zeta_k^{(t)} \|\mathbf{w}_{t+1} - \widehat{\mathbf{w}}_{k+1}\|^2$, $n_{t+1} = \sum_{k=0}^{t} \zeta_k^{(t)} [\nabla f(\widehat{\mathbf{w}}_{k+1}; \xi_{k+1}) - F(\widehat{\mathbf{w}}_{k+1})]$, where $\nabla f(\widehat{\mathbf{w}}_{k+1}; \xi_{k+1})$ is an unbiased stochastic first-order oracle for $F(\widehat{\mathbf{w}}_{k+1})$ with bounded variance $\sigma^2$. Note that $\nabla F$ is a $L_H$-smooth mapping (according to Assumption 1), then by Lemma 10 of (Wang et al., 2017), we have

$$\|\mathbf{z}_t - \nabla F(\mathbf{w}_t)\|^{3/2} \leq (L_H m_t + \|n_t\|)^{3/2} \leq 2L_H^{3/2} m_t^{3/2} + 2\|n_t\|^{3/2}.$$

Define $q_{t+1} = \sum_{k=0}^{t} \zeta_k^{(t)} \|\mathbf{w}_{t+1} - \widehat{\mathbf{w}}_{k+1}\|$. According to Lemma 11 (a) and (b) of (Wang et al., 2017), we have

$$m_{t+1} + 4q_{t+1}^2 \leq \left(1 - \frac{\beta}{2}\right)(m_t + 4q_t^2) + \frac{18}{\beta}\|\mathbf{w}_{t+1} - \mathbf{w}_t\|^2.$$

Taking the power $3/2$ on both sides of the inequality and using the fact that $(a+b)^{3/2} \leq \sqrt{1+\frac{\beta}{2}}a^{3/2} + \sqrt{1+\frac{2}{\beta}}b^{3/2}$ for $\beta > 0$, we have

$$\begin{aligned} &\left(m_{t+1} + 4q_{t+1}^2\right)^{3/2} \\ &\leq \left(1 + \frac{\beta}{2}\right)^{1/2}\left(1 - \frac{\beta}{2}\right)^{3/2}(m_t + 4q_t^2)^{3/2} + \left(1 + \frac{2}{\beta}\right)^{1/2}\frac{80}{\beta^{3/2}}\|\mathbf{w}_{t+1} - \mathbf{w}_t\|^3 \\ &\leq \left(1 - \frac{\beta}{2}\right)(m_t + 4q_t^2)^{3/2} + \frac{160}{\beta^2}\|\mathbf{w}_{t+1} - \mathbf{w}_t\|^3, \end{aligned} \tag{16}$$

where the last inequality holds since $1/\beta \geq 1$.

By the definition of $n_t$, we have $n_{t+1} = (1-\beta)n_t + \beta(\nabla f(\widehat{\mathbf{w}}_{t+1}) - F(\widehat{\mathbf{w}}_{t+1}))$. Denote $\mathcal{F}_{t+1}$ by the $\sigma$-algebra generated by $\xi_1, \ldots, \xi_{t+1}$. Noting that

$$\mathbb{E}\left[\|n_{t+1}\|^{3/2}|\mathcal{F}_{t+1}\right] \leq \left(\mathbb{E}\left[\|n_{t+1}\|^2|\mathcal{F}_{t+1}\right]\right)^{3/4} \leq (1-\beta/2)^{3/2}\|n_t\|^{3/2} + \beta^{3/2}\sigma^{3/2}, \tag{17}$$

where the last inequality holds by invoking Lemma 11(c) of (Wang et al., 2017). Define $\delta_t^{3/2} = 2L_H^{3/2}(m_t + 4q_t^2)^{3/2} + 2\|n_t\|^{3/2}$, then we have $\|\mathbf{z}_t - \nabla F(\mathbf{w}_t)\|^{3/2} \leq \delta_t^{3/2}$ for all $t$. According to (16) and (17), we have

$$\mathbb{E}\left[\delta_{t+1}^{3/2}|\mathcal{F}_{t+1}\right] \leq \left(1 - \frac{\beta}{2}\right)\|\delta_t\|^{3/2} + 2\beta^{3/2}\sigma^{3/2} + \frac{320L_H^{3/2}\|\mathbf{w}_{t+1} - \mathbf{w}_t\|^3}{\beta^2}.$$

Taking expectation on both sides yields

$$\mathbb{E}\left[\delta_{t+1}^{3/2}\right] \leq \left(1 - \frac{\beta}{2}\right)\mathbb{E}\left[\delta_t^{3/2}\right] + 2\beta^{3/2}\sigma^{3/2} + \mathbb{E}\left[\frac{320L_H^{3/2}\|\mathbf{w}_{t+1} - \mathbf{w}_t\|^3}{\beta^2}\right]. \qquad \square$$

**Lemma 4.** *Adam$^+$ with learning rate $\eta_t = \frac{\alpha\beta}{\max(\|\mathbf{z}_t\|^{1/2}, \epsilon_0)}$ and $640\alpha^3 L_H^{3/2} \leq 1/120$ satisfies*

$$\frac{1}{T}\sum_{t=1}^{T} \mathbb{E}\left[\|\nabla F(\mathbf{w}_t)\|^{3/2}\right] \leq \frac{101\Delta}{\alpha\beta T} + \frac{2727\mathbb{E}\left[\delta_1^{3/2}\right]}{\beta T} + 4545\beta^{1/2}\sigma^{3/2} + \frac{3\beta^3 L^{3/2}}{100}.$$

*To ensure that $\frac{1}{T}\sum_{t=1}^{T} \mathbb{E}\left[\|\nabla F(\mathbf{w}_t)\|^{3/2}\right] \leq \epsilon^{3/2}$, we can choose $\beta = \epsilon^3$, $T = O(\epsilon^{-9/2})$.*

*Proof.* By Lemma 3 and noting that $\eta_t = \frac{\alpha\beta}{\max\left(\|\mathbf{z}_t\|^{1/2}, \epsilon_0\right)}$, we have

$$
\begin{aligned}
\mathbb{E}\left[\delta_{t+1}^{3/2}\right] &\leq \left(1 - \frac{\beta}{2}\right)\mathbb{E}\left[\delta_t^{3/2}\right] + 2\beta^{3/2}\sigma^{3/2} + \mathbb{E}\left[\frac{320 L_H^{3/2}\alpha^3\beta^3\|\mathbf{z}_t\|^{3/2}}{\beta^2}\right] \\
&\leq \left(1 - \frac{\beta}{2}\right)\mathbb{E}\left[\delta_t^{3/2}\right] + 2\beta^{3/2}\sigma^{3/2} + \mathbb{E}\left[640 L_H^{3/2}\alpha^3\beta\left(\|\nabla F(\mathbf{w}_t)\|^{3/2} + \|\delta_t\|^{3/2}\right)\right].
\end{aligned}
\tag{18}
$$

Note that $640\alpha^3 L_H^{3/2} \leq 1/120$. Plugging it into (18), we have

$$
\frac{59\beta}{120}\mathbb{E}\left[\delta_t^{3/2}\right] \leq \mathbb{E}\left[\delta_t^{3/2} - \delta_{t+1}^{3/2}\right] + 2\beta^{3/2}\sigma^{3/2} + \mathbb{E}\left[\frac{\beta}{120}\|\nabla F(\mathbf{w}_t)\|^{3/2}\right].
\tag{19}
$$

Summing over $t = 1, \ldots, T$ on both sides of (19) and with some simple algebra, we have

$$
\sum_{t=1}^{T}\mathbb{E}\left[\delta_t^{3/2}\right] \leq \sum_{t=1}^{T}\mathbb{E}\left[\frac{3(\delta_t^{3/2} - \delta_{t+1}^{3/2})}{\beta}\right] + \sum_{t=1}^{T} 5\beta^{1/2}\sigma^{3/2} + \sum_{t=1}^{T}\mathbb{E}\left[\frac{1}{59}\|\nabla F(\mathbf{w}_t)\|^{2/3}\right].
$$

By Lemma 2, taking expectation on both sides, we have

$$
\mathbb{E}\left[F(\mathbf{w}_{t+1}) - F(\mathbf{w}_t)\right] \leq \alpha\beta\left(-\frac{\mathbb{E}\left[\|\nabla F(\mathbf{w}_t)\|^{3/2}\right]}{6} + 9\mathbb{E}\left[\delta_t^{3/2}\right]\right) + \frac{64\alpha^4\beta^4 L^3}{3}.
\tag{20}
$$

Summing (20) over $t = 1, \ldots, T$ yields

$$
\frac{5}{504}\alpha\beta\sum_{t=1}^{T}\mathbb{E}\left[\|\nabla F(\mathbf{w}_t)\|^{3/2}\right] \leq F(\mathbf{w}_1) - F_* + \alpha\beta\left(\frac{27\mathbb{E}\left[\delta_1^{3/2}\right]}{\beta} + \sum_{t=1}^{T} 45\beta^{1/2}\sigma^{3/2}\right) + \frac{64\alpha^4\beta^4 L^3 T}{3}.
$$

Hence, we have

$$
\begin{aligned}
\frac{1}{T}\sum_{t=1}^{T}\mathbb{E}\left[\|\nabla F(\mathbf{w}_t)\|^{3/2}\right] &\leq \frac{101\Delta}{\alpha\beta T} + \frac{2727\mathbb{E}\left[\delta_1^{3/2}\right]}{\beta} + 4545\beta^{1/2}\sigma^{3/2} + 2155\alpha^3\beta^3 L^3 \\
&\leq \frac{101\Delta}{\alpha\beta T} + \frac{2727\mathbb{E}\left[\delta_1^{3/2}\right]}{\beta T} + 4545\beta^{1/2}\sigma^{3/2} + \frac{3\beta^3 L^{3/2}}{100}. \qquad \square
\end{aligned}
$$

**Lemma 5.** *Under the same setting of Lemma 4, we know that to ensure that $\frac{1}{T}\sum_{t=1}^{T}\mathbb{E}\left[\delta_t^{3/2}\right] \leq \epsilon^{3/2}$, we need $T = O(\epsilon^{-9/2})$ iterations.*

*Proof.* From (19) and Lemma 4, we have

$$
\sum_{t=1}^{T}\frac{59\beta}{120}\mathbb{E}\left[\delta_t^{3/2}\right] \leq \mathbb{E}\left[\delta_1^{3/2}\right] + 2\beta^{3/2}\sigma^{3/2}T + \sum_{t=1}^{T}\mathbb{E}\left[\frac{\beta}{120}\|\nabla F(\mathbf{w}_t)\|^{3/2}\right].
$$

Noting that $\beta = T^{-b}$ with $0 < b < 1$, then we know that there exists a universal constant $C > 0$ such that

$$
\frac{1}{T}\sum_{t=1}^{T}\frac{59}{120}\mathbb{E}\left[\delta_t^{3/2}\right] \leq \frac{\mathbb{E}\left[\delta_1^{3/2}\right]}{T^{1-b}} + \frac{2\sigma^{3/2}}{T^{b/2}} + \frac{1}{T}\sum_{t=1}^{T}\mathbb{E}\left[\frac{1}{120}\|\nabla F(\mathbf{w}_t)\|^{3/2}\right].
\tag{21}
$$

Take $b = \frac{2}{3}$. From Lemma 4, we know that it takes $T = O(\epsilon^{-9/2})$ iterations to ensure that $\frac{1}{T}\sum_{t=1}^{T}\mathbb{E}\left[\|\nabla F(\mathbf{w}_t)\|^{3/2}\right] \leq \epsilon^{3/2}$. In addition, From (21), we know that it takes $T = O(\epsilon^{-9/2})$ iterations to ensure that $\frac{1}{T}\sum_{t=1}^{T}\mathbb{E}\left[\delta_t^{3/2}\right] \leq \epsilon^{3/2}$. $\qquad\square$

We can easily prove Theorem 2 by incorporating the results in Lemma 4 and Lemma 5. It is also evident to see that if $\beta = 1/T^s$ with $0 < s < 1$, then it takes $T = O\left(\text{poly}(1/\epsilon)\right)$ number of iterations to ensure that $\frac{1}{T}\sum_{t=1}^{T}\mathbb{E}\left[\delta_t^{3/2}\right] \leq \epsilon^{3/2}$ and $\frac{1}{T}\sum_{t=1}^{T}\mathbb{E}\left[\|\nabla F(\mathbf{w}_t)\|^{3/2}\right] \leq \epsilon^{3/2}$ hold simultaneously.

# D    PROOF OF THEOREM 3

*Proof.* Define $\gamma_t = \min\left(\frac{\beta^a}{\|\mathbf{z}_t\|^{2/3}}, \frac{\beta^a}{\epsilon_0}\right)$ with $\epsilon_0 = 2\beta^a$. Then we know that $\eta_t = \alpha\gamma_t$ and $\gamma_t \leq \frac{1}{2}$. Note that $\alpha \leq \frac{1}{L}$, so we have $\eta_t \leq \frac{1}{2L}$. By the $L$-smoothness of $F$, we have

$$F(\mathbf{w}_{t+1}) \leq F(\mathbf{w}_t) + \nabla^\top F(\mathbf{w}_t)(\mathbf{w}_{t+1} - \mathbf{w}_t) + \frac{L}{2}\|\mathbf{w}_{t+1} - \mathbf{w}_t\|^2$$

$$\leq F(\mathbf{w}_t) - \eta_t\nabla^\top F(\mathbf{w}_t)\mathbf{z}_t + \left(\frac{\eta_t^2 L}{2} + \frac{\gamma_t}{2L}\right)\|\mathbf{z}_t\|^2 - \frac{1}{2L}\gamma_t\|\mathbf{z}_t\|^2$$

$$= F(\mathbf{w}_t) - \eta_t\nabla^\top F(\mathbf{w}_t)(\mathbf{z}_t - \nabla F(\mathbf{w}_t) + \nabla F(\mathbf{w}_t)) + \left(\frac{\eta_t^2 L}{2} + \frac{\gamma_t}{2L}\right)\|\mathbf{z}_t\|^2 - \frac{1}{2L}\gamma_t\|\mathbf{z}_t\|^2$$

$$\overset{(a)}{\leq} F(\mathbf{w}_t) - \eta_t\nabla^\top F(\mathbf{w}_t)(\mathbf{z}_t - \nabla F(\mathbf{w}_t) + \nabla F(\mathbf{w}_t)) + \left(\eta_t^2 L + \frac{\gamma_t}{L}\right)\left(\|\mathbf{z}_t - \nabla F(\mathbf{w}_t)\|^2 + \|\nabla F(\mathbf{w}_t)\|^2\right) - \frac{1}{2L}\gamma_t\|\mathbf{z}_t\|^2$$

$$\overset{(b)}{\leq} F(\mathbf{w}_t) - \frac{\eta_t}{2}\|\nabla F(\mathbf{w}_t)\|^2 + \frac{\eta_t}{2}\|\mathbf{z}_t - \nabla F(\mathbf{w}_t)\|^2 + \left(\eta_t^2 L + \frac{\gamma_t}{L}\right)\left(\|\mathbf{z}_t - \nabla F(\mathbf{w}_t)\|^2 + \|\nabla F(\mathbf{w}_t)\|^2\right) - \frac{1}{2L}\gamma_t\|\mathbf{z}_t\|^2$$

$$= F(\mathbf{w}_t) - \left(\frac{\eta_t}{2} - \eta_t^2 L - \frac{\gamma_t}{L}\right)\|\nabla F(\mathbf{w}_t)\|^2 + \left(\eta_t^2 L + \frac{\gamma_t}{L} + \frac{\eta_t}{2}\right)\|\mathbf{z}_t - \nabla F(\mathbf{w}_t)\|^2 - \frac{1}{2L}\gamma_t\|\mathbf{z}_t\|^2$$

$$\overset{(c)}{\leq} F(\mathbf{w}_t) - \frac{1}{2L}\gamma_t\|\mathbf{z}_t\|^2 + \frac{1}{L}\|\mathbf{z}_t - \nabla F(\mathbf{w}_t)\|^2,$$

$$(22)$$

where (a) holds since $\|\mathbf{z}_t\|^2 \leq 2\|\mathbf{z}_t - \nabla F(\mathbf{w}_t)\|^2 + 2\|\nabla F(\mathbf{w}_t)\|^2$, (b) holds since $-\nabla^\top F(\mathbf{w}_t)\mathbf{z}_t \leq \frac{1}{2}\left(\|\nabla F(\mathbf{w}_t)\|^2 + \|\mathbf{z}_t - \nabla F(\mathbf{w}_t)\|^2\right)$, (c) holds due to $\frac{\eta_t}{2} - \eta_t^2 L - \frac{\gamma_t}{L} \geq 0$ (since $\eta_t \leq \frac{1}{2L}$, we have $\frac{\eta_t}{2} - \eta_t^2 L \geq \frac{\gamma_t}{2L}$ and note that $\frac{\gamma_t}{L} \leq \frac{1}{2L}$).

By the definition of $\gamma_t$, we have

$$\gamma_t\|\mathbf{z}_t\|^2 \geq \beta^{2a}\|\mathbf{z}_t\|^{2/3}\min\left(\frac{\|\mathbf{z}_t\|^{2/3}}{\beta^a}, \frac{\|\mathbf{z}_t\|^{4/3}}{\beta^a\epsilon_0}\right) = \beta^{2a}\|\mathbf{z}_t\|^{2/3}\min\left(\frac{\|\mathbf{z}_t\|^{2/3}}{\beta^a}, \frac{\|\mathbf{z}_t\|^{4/3}}{2\beta^{2a}}\right)$$

$$\overset{(a)}{\geq} \beta^{2a}\|\mathbf{z}_t\|^{2/3}\left(\frac{\|\mathbf{z}_t\|^{2/3}}{\beta^a} - \frac{1}{2}\right) = \beta^a\|\mathbf{z}_t\|^{4/3} - \frac{\beta^{2a}\|\mathbf{z}_t\|^{2/3}}{2}, \qquad (23)$$

where (a) holds since $x \geq x - \frac{1}{2}$, $\frac{x^2}{2} \geq x - \frac{1}{2}$ hold for any $x$ and let $x = \frac{\|\mathbf{z}_t\|^{2/3}}{\beta^a}$.

Combining (22) and (23), we have

$$\beta^a\|\mathbf{z}_t\|^{4/3} \leq \gamma_t\|\mathbf{z}_t\|^2 + \frac{\beta^{2a}\|\mathbf{z}_t\|^{2/3}}{2} \leq 2L\left(F(\mathbf{w}_t) - F(\mathbf{w}_{t+1})\right) + \frac{\beta^{2a}\|\mathbf{z}_t\|^{2/3}}{2} + 2\|\mathbf{z}_t - \nabla F(\mathbf{w}_t)\|^2$$

$$= 2L\left(F(\mathbf{w}_t) - F(\mathbf{w}_{t+1})\right) + \beta^a\|\mathbf{z}_t\|^{4/3} \cdot \frac{\beta^a}{2\|\mathbf{z}_t\|^{2/3}} + 2\|\mathbf{z}_t - \nabla F(\mathbf{w}_t)\|^2.$$

If $\frac{\beta^a}{2\|\mathbf{z}_t\|^{2/3}} \leq \frac{1}{2}$, we have $\beta^a\|\mathbf{z}_t\|^{4/3} \leq 4L\left(F(\mathbf{w}_t) - F(\mathbf{w}_{t+1})\right) + 4\|\mathbf{z}_t - \nabla F(\mathbf{w}_t)\|^2$. If $\frac{\beta^a}{2\|\mathbf{z}_t\|^{2/3}} > \frac{1}{2}$, then $\beta^a > \|\mathbf{z}_t\|^{2/3}$, and hence we have $\beta^a\|\mathbf{z}_t\|^{4/3} \leq \beta^{3a}$. As a result, we have

$$\beta^a\|\mathbf{z}_t\|^{4/3} \leq 4L\left(F(\mathbf{w}_t) - F(\mathbf{w}_{t+1})\right) + 4\|\mathbf{z}_t - \nabla F(\mathbf{w}_t)\|^2 + \beta^{3a}. \qquad (24)$$

Taking summation on both sides of (24) over $t = 1, \ldots, T$ yields

$$\sum_{t=1}^T \|\mathbf{z}_t\|^{4/3} \leq 4L\sum_{t=1}^T \frac{F(\mathbf{w}_t) - F(\mathbf{w}_{t+1})}{\beta^a} + \sum_{t=1}^T \frac{4}{\beta^a}\|\mathbf{z}_t - \nabla F(\mathbf{w}_t)\|^2 + \beta^{2a}T. \qquad (25)$$

Define $\Delta_t = \mathbf{z}_t - \nabla F(\mathbf{w}_t)$, then we have

$$\|\nabla F(\mathbf{w}_t)\|^{4/3} \leq 2\|\mathbf{z}_t\|^{4/3} + 2\|\Delta_t\|^{4/3}. \qquad (26)$$

Hence,

$$
\sum_{t=1}^{T} \|\mathbf{z}_t\|^{4/3} + \|\nabla F(\mathbf{w}_t)\|^{4/3}
$$

$$
\overset{(a)}{\leq} 2\sum_{t=1}^{T} \|\Delta_t\|^{4/3} + 12L\sum_{t=1}^{T} \frac{F(\mathbf{w}_t) - F(\mathbf{w}_{t+1})}{\beta^a} + \sum_{t=1}^{T} \frac{12}{\beta^a}\|\mathbf{z}_t - \nabla F(\mathbf{w}_t)\|^2 + 3\beta^{2a}T
$$

$$
\overset{(b)}{\leq} \sum_{t=1}^{T} \frac{4}{3}\left(\frac{\|\Delta_t\|^2}{\beta^a} + \frac{\beta^{2a}}{2}\right) + 12L\sum_{t=1}^{T} \frac{F(\mathbf{w}_t) - F(\mathbf{w}_{t+1})}{\beta^a} + \sum_{t=1}^{T} \frac{12}{\beta^a}\|\mathbf{z}_t - \nabla F(\mathbf{w}_t)\|^2 + 3\beta^{2a}T
$$

$$
\leq 12L\sum_{t=1}^{T} \frac{F(\mathbf{w}_t) - F(\mathbf{w}_{t+1})}{\beta^a} + \sum_{t=1}^{T} \frac{14}{\beta^a}\|\mathbf{z}_t - \nabla F(\mathbf{w}_t)\|^2 + 4\beta^{2a}T,
$$

$$(27)$$

where (a) holds due to (25) and (26), (b) holds because $\min_{x>0} \frac{c^2}{x} + \frac{x^2}{2} = \frac{3c^{4/3}}{2}$.

By Lemma 1, we know that

$$
\mathbb{E}\left[\delta_{t+1}^2\right] \leq \left(1 - \frac{\beta}{2}\right)\mathbb{E}\left[\delta_t^2\right] + 2\beta^2\sigma^2 + \mathbb{E}\left[\frac{CL^2\eta_t^4\|\mathbf{z}_t\|^4}{\beta^3}\right]
$$

$$
\overset{(a)}{\leq} \left(1 - \frac{\beta}{2}\right)\mathbb{E}\left[\delta_t^2\right] + 2\beta^2\sigma^2 + \mathbb{E}\left[\frac{CL^2\alpha^4\beta^{4a}\|\mathbf{z}_t\|^4}{\max(\|\mathbf{z}_t\|^{8/3}, \epsilon_0^4)\beta^3}\right]
$$

$$
\leq \left(1 - \frac{\beta}{2}\right)\mathbb{E}\left[\delta_t^2\right] + 2\beta^2\sigma^2 + \mathbb{E}\left[CL^2\alpha^4\beta^{4a-3}\|\mathbf{z}_t\|^{4/3}\right].
$$

Note that $CL^2\alpha^4 \leq 1/14$, we have

$$
\frac{\beta}{2}\mathbb{E}\left[\delta_t^2\right] \leq \mathbb{E}\left[\delta_t^2 - \delta_{t+1}^2\right] + 2\beta^2\sigma^2 + \mathbb{E}\left[\frac{\beta^{4a-3}\|\mathbf{z}_t\|^{4/3}}{14}\right]. \qquad (28)
$$

Taking summation on both sides of (28) over $t = 1, \ldots, T$, we have

$$
\sum_{t=1}^{T} \mathbb{E}\left[\delta_t^2\right] \leq \frac{\mathbb{E}\left[\delta_1^2\right]}{\beta} + 2\beta\sigma^2 T + \sum_{t=1}^{T} \mathbb{E}\left[\frac{\beta^{4a-4}\|\mathbf{z}_t\|^{4/3}}{14}\right]
$$

$$
= \frac{\mathbb{E}\left[\delta_1^2\right]}{\beta} + 2\beta\sigma^2 T + \sum_{t=1}^{T} \mathbb{E}\left[\frac{\beta^a\|\mathbf{z}_t\|^{4/3}}{14}\right], \qquad (29)
$$

where the last equality holds since $a = 4/3$.

Taking expectation on both sides of (27) and combining (29), we have

$$
\sum_{t=1}^{T} \mathbb{E}\left[\|\mathbf{z}_t\|^{4/3} + \|\nabla F(\mathbf{w}_t)\|^{4/3}\right] \leq \frac{12L\Delta}{\beta^a} + \frac{14\mathbb{E}\left[\delta_1^2\right]}{\beta^{1+a}} + \frac{28\beta\sigma^2 T}{\beta^a} + \sum_{t=1}^{T} \mathbb{E}\left[\|\mathbf{z}_t\|^{4/3}\right] + 4\beta^{2a}T.
$$

As a result, we have

$$
\frac{1}{T}\sum_{t=1}^{T} \mathbb{E}\left[\|\nabla F(\mathbf{w}_t)\|^{4/3}\right] \leq \frac{12L\Delta}{\beta^a T} + \frac{14\mathbb{E}\left[\delta_1^2\right]}{\beta^{1+a}T} + \frac{28\beta\sigma^2}{\beta^a} + 4\beta^{2a}.
$$

Suppose initial batch size is $T_0$, the intermediate batch size is $m$, and $a = 4/3$, then we have

$$
\frac{1}{T}\sum_{t=1}^{T} \mathbb{E}\left[\|\nabla F(\mathbf{w}_t)\|^{4/3}\right] \leq \frac{12L\Delta}{\beta^{4/3}T} + \frac{14\sigma^2}{\beta^{7/3}T_0 T} + \frac{28\sigma^2}{\beta^{1/3}m} + 4\beta^{8/3}. \qquad (30)
$$

We can choose $\beta = O(\epsilon^{1/2})$, $T = O(\epsilon^{-2})$, the initial batch size $T_0 = 1/\beta = O(\epsilon^{-1/2})$, the intermediate batch size as $m = 1/\beta^3 = O(\epsilon^{-3/2})$, which ends up with the total complexity $O(\epsilon^{-3.5})$. $\quad\square$

## E  A New Variant of Adam$^+$

**Theorem 4.** *Assume that $\|\nabla f(\mathbf{w};\xi)\| \leq G$ almost surely for every $\mathbf{w} \in \mathbb{R}^d$. Choose $\eta_t = \frac{\alpha\beta^a}{\max\left(\|\mathbf{z}_t\|^{1/2},\epsilon_0\right)}$ with $a = 4/3$, and we have*

$$\frac{1}{T}\sum_{t=1}^{T}\mathbb{E}\left[\|\nabla F(\mathbf{w}_t)\|^{3/2}\right] \leq \frac{12L\Delta}{\beta^a T} + \frac{14\mathbb{E}\left[\delta_1^2\right]}{\beta^{1+a}T} + \frac{28\beta\sigma^2}{\beta^a} + 4\beta^{3a}.$$

*Denote the initial batch size and the intermediate batch size are $T_0$ and $m$ respectively, then we have*

$$\frac{1}{T}\sum_{t=1}^{T}\mathbb{E}\left[\|\nabla F(\mathbf{w}_t)\|^{3/2}\right] \leq \frac{12L\Delta}{\beta^a T} + \frac{14\sigma^2}{T_0 T \beta^{1+a}} + \frac{28\sigma^2}{\beta^{a-1}m} + 4\beta^{3a}.$$

*To ensure that $\frac{1}{T}\sum_{t=1}^{T}\mathbb{E}\left[\|\nabla F(\mathbf{w}_t)\|^{3/2}\right] \leq \epsilon^{3/2}$, we choose $\beta = \epsilon^{3/8}$, $T = O(1/\epsilon^2)$, the initial batch size is $T_0 = 1/\epsilon^{3/8}$ and $m = 1/\epsilon^{1.625}$, then the total computational complexity is $O(1/\epsilon^{3.625})$.*

*Proof.* Define $\gamma_t = \min\left(\frac{\beta^a}{\|\mathbf{z}_t\|^{1/2}}, \frac{\beta^a}{\epsilon_0}\right)$ with $\epsilon_0 = 2\beta^a$. Then we know that $\eta_t = \alpha\gamma_t$ and $\gamma_t \leq \frac{1}{2}$. Note that $\alpha \leq \frac{1}{L}$, so we have $\eta_t \leq \frac{1}{2L}$. By the $L$-smoothness of $F$, we have

$$F(\mathbf{w}_{t+1}) \leq F(\mathbf{w}_t) + \nabla^\top F(\mathbf{w}_t)(\mathbf{w}_{t+1} - \mathbf{w}_t) + \frac{L}{2}\|\mathbf{w}_{t+1} - \mathbf{w}_t\|^2$$

$$\leq F(\mathbf{w}_t) - \eta_t \nabla^\top F(\mathbf{w}_t)\mathbf{z}_t + \left(\frac{\eta_t^2 L}{2} + \frac{\gamma_t}{2L}\right)\|\mathbf{z}_t\|^2 - \frac{1}{2L}\gamma_t\|\mathbf{z}_t\|^2$$

$$= F(\mathbf{w}_t) - \eta_t\nabla^\top F(\mathbf{w}_t)\left(\mathbf{z}_t - \nabla F(\mathbf{w}_t) + \nabla F(\mathbf{w}_t)\right) + \left(\frac{\eta_t^2 L}{2} + \frac{\gamma_t}{2L}\right)\|\mathbf{z}_t\|^2 - \frac{1}{2L}\gamma_t\|\mathbf{z}_t\|^2$$

$$\overset{(a)}{\leq} F(\mathbf{w}_t) - \eta_t\nabla^\top F(\mathbf{w}_t)\left(\mathbf{z}_t - \nabla F(\mathbf{w}_t) + \nabla F(\mathbf{w}_t)\right) + \left(\eta_t^2 L + \frac{\gamma_t}{L}\right)\left(\|\mathbf{z}_t - \nabla F(\mathbf{w}_t)\|^2 + \|\nabla F(\mathbf{w}_t)\|^2\right) - \frac{1}{2L}\gamma_t\|\mathbf{z}_t\|^2$$

$$\overset{(b)}{\leq} F(\mathbf{w}_t) - \frac{\eta_t}{2}\|\nabla F(\mathbf{w}_t)\|^2 + \frac{\eta_t}{2}\|\mathbf{z}_t - \nabla F(\mathbf{w}_t)\|^2 + \left(\eta_t^2 L + \frac{\gamma_t}{L}\right)\left(\|\mathbf{z}_t - \nabla F(\mathbf{w}_t)\|^2 + \|\nabla F(\mathbf{w}_t)\|^2\right) - \frac{1}{2L}\gamma_t\|\mathbf{z}_t\|^2$$

$$= F(\mathbf{w}_t) - \left(\frac{\eta_t}{2} - \eta_t^2 L - \frac{\gamma_t}{L}\right)\|\nabla F(\mathbf{w}_t)\|^2 + \left(\eta_t^2 L + \frac{\gamma_t}{L} + \frac{\eta_t}{2}\right)\|\mathbf{z}_t - \nabla F(\mathbf{w}_t)\|^2 - \frac{1}{2L}\gamma_t\|\mathbf{z}_t\|^2$$

$$\overset{(c)}{\leq} F(\mathbf{w}_t) - \frac{1}{2L}\gamma_t\|\mathbf{z}_t\|^2 + \frac{1}{L}\|\mathbf{z}_t - \nabla F(\mathbf{w}_t)\|^2,$$

(31)

where (a) holds since $\|\mathbf{z}_t\|^2 \leq 2\|\mathbf{z}_t - \nabla F(\mathbf{w}_t)\|^2 + 2\|\nabla F(\mathbf{w}_t)\|^2$, (b) holds since $-\nabla^\top F(\mathbf{w}_t)\mathbf{z}_t \leq \frac{1}{2}\left(\|\nabla F(\mathbf{w}_t)\|^2 + \|\mathbf{z}_t - \nabla F(\mathbf{w}_t)\|^2\right)$, (c) holds due to $\frac{\eta_t}{2} - \eta_t^2 L - \frac{\gamma_t}{L} \geq 0$ (since $\eta_t \leq \frac{1}{2L}$, we have $\frac{\eta_t}{2} - \eta_t^2 L \geq \frac{1}{2L}$ and note that $\frac{\gamma_t}{L} \leq \frac{1}{2L}$).

By the definition of $\gamma_t$, we have

$$\gamma_t\|\mathbf{z}_t\|^2 \geq \beta^{2a}\|\mathbf{z}_t\|\min\left(\frac{\|\mathbf{z}_t\|^{1/2}}{\beta^a}, \frac{\|\mathbf{z}_t\|}{\beta^a\epsilon_0}\right) = \beta^{2a}\|\mathbf{z}_t\|\min\left(\frac{\|\mathbf{z}_t\|^{1/2}}{\beta^a}, \frac{\|\mathbf{z}_t\|}{2\beta^{2a}}\right)$$

$$\overset{(a)}{\geq} \beta^{2a}\|\mathbf{z}_t\|\left(\frac{\|\mathbf{z}_t\|^{1/2}}{\beta^a} - \frac{1}{2}\right) = \beta^a\|\mathbf{z}_t\|^{3/2} - \frac{\beta^{2a}\|\mathbf{z}_t\|}{2},$$

(32)

where (a) holds since $x \geq x - \frac{1}{2}$, $\frac{x^2}{2} \geq x - \frac{1}{2}$ hold for any $x$ and let $x = \frac{\|\mathbf{z}_t\|^{1/2}}{\beta^a}$.

Combining (31) and (32), we have

$$\beta^a\|\mathbf{z}_t\|^{3/2} \leq \gamma_t\|\mathbf{z}_t\|^2 + \frac{\beta^{2a}\|\mathbf{z}_t\|}{2} \leq 2L\left(F(\mathbf{w}_t) - F(\mathbf{w}_{t+1})\right) + \frac{\beta^{2a}\|\mathbf{z}_t\|}{2} + 2\|\mathbf{z}_t - \nabla F(\mathbf{w}_t)\|^2$$

$$= 2L\left(F(\mathbf{w}_t) - F(\mathbf{w}_{t+1})\right) + \beta^a\|\mathbf{z}_t\|^{3/2}\cdot\frac{\beta^a}{2\|\mathbf{z}_t\|^{1/2}} + 2\|\mathbf{z}_t - \nabla F(\mathbf{w}_t)\|^2.$$

If $\frac{\beta^a}{2\|\mathbf{z}_t\|^{1/2}} \leq \frac{1}{2}$, we have $\beta^a \|\mathbf{z}_t\|^{4/3} \leq 4L \left( F(\mathbf{w}_t) - F(\mathbf{w}_{t+1}) \right) + 4\|\mathbf{z}_t - \nabla F(\mathbf{w}_t)\|^2$. If $\frac{\beta^a}{2\|\mathbf{z}_t\|^{1/2}} > \frac{1}{2}$, then $\beta^a > \|\mathbf{z}_t\|^{1/2}$, and hence we have $\beta^a \|\mathbf{z}_t\|^{3/2} \leq \beta^{4a}$. As a result, we have

$$\beta^a \|\mathbf{z}_t\|^{3/2} \leq 4L \left( F(\mathbf{w}_t) - F(\mathbf{w}_{t+1}) \right) + 4\|\mathbf{z}_t - \nabla F(\mathbf{w}_t)\|^2 + \beta^{4a}. \tag{33}$$

Taking summation on both sides of (33) over $t = 1, \ldots, T$ yields

$$\sum_{t=1}^{T} \|\mathbf{z}_t\|^{3/2} \leq 4L \sum_{t=1}^{T} \frac{F(\mathbf{w}_t) - F(\mathbf{w}_{t+1})}{\beta^a} + \sum_{t=1}^{T} \frac{4}{\beta^a} \|\mathbf{z}_t - \nabla F(\mathbf{w}_t)\|^2 + \beta^{3a} T. \tag{34}$$

Define $\Delta_t = \mathbf{z}_t - \nabla F(\mathbf{w}_t)$, then we have

$$\|\nabla F(\mathbf{w}_t)\|^{3/2} \leq 2\|\mathbf{z}_t\|^{3/2} + 2\|\Delta_t\|^{3/2}. \tag{35}$$

Hence, we have

$$\sum_{t=1}^{T} \|\mathbf{z}_t\|^{3/2} + \|\nabla F(\mathbf{w}_t)\|^{3/2}$$

$$\overset{(a)}{\leq} 2\sum_{t=1}^{T} \|\Delta_t\|^{3/2} + 12L \sum_{t=1}^{T} \frac{F(\mathbf{w}_t) - F(\mathbf{w}_{t+1})}{\beta^a} + \sum_{t=1}^{T} \frac{12}{\beta^a} \|\mathbf{z}_t - \nabla F(\mathbf{w}_t)\|^2 + 3\beta^{3a} T$$

$$\overset{(b)}{\leq} \sum_{t=1}^{T} \frac{3}{2} \left( \frac{\|\Delta_t\|^2}{\beta^a} + \frac{\beta^{3a}}{3} \right) + 12L \sum_{t=1}^{T} \frac{F(\mathbf{w}_t) - F(\mathbf{w}_{t+1})}{\beta^a} + \sum_{t=1}^{T} \frac{12}{\beta^a} \|\mathbf{z}_t - \nabla F(\mathbf{w}_t)\|^2 + 3\beta^{3a} T$$

$$\leq 12L \sum_{t=1}^{T} \frac{F(\mathbf{w}_t) - F(\mathbf{w}_{t+1})}{\beta^a} + \sum_{t=1}^{T} \frac{14}{\beta^a} \|\mathbf{z}_t - \nabla F(\mathbf{w}_t)\|^2 + 4\beta^{3a} T, \tag{36}$$

where (a) holds due to (34) and (35), (b) holds because $\min_{x>0} \frac{c^2}{x} + \frac{x^3}{3} = \frac{4c^{3/2}}{3}$.

By Lemma 1, we know that

$$\mathbb{E}\left[\delta_{t+1}^2\right] \leq \left(1 - \frac{\beta}{2}\right) \mathbb{E}\left[\delta_t^2\right] + 2\beta^2 \sigma^2 + \mathbb{E}\left[\frac{CL_H^2 \eta_t^4 \|\mathbf{z}_t\|^4}{\beta^3}\right]$$

$$\overset{(a)}{\leq} \left(1 - \frac{\beta}{2}\right) \mathbb{E}\left[\delta_t^2\right] + 2\beta^2 \sigma^2 + \mathbb{E}\left[\frac{CL_H^2 \alpha^4 \beta^{4a} \|\mathbf{z}_t\|^4}{\max(\|\mathbf{z}_t\|^2, \epsilon_0^4)\beta^3}\right]$$

$$\leq \left(1 - \frac{\beta}{2}\right) \mathbb{E}\left[\delta_t^2\right] + 2\beta^2 \sigma^2 + \mathbb{E}\left[CL_H^2 \alpha^4 \beta^{4a-3} \|\mathbf{z}_t\|^2\right].$$

Note that $CL_H^2 \alpha^4 \leq \frac{1}{14G^{1/2}}$, we have

$$\frac{\beta}{2} \mathbb{E}\left[\delta_t^2\right] \leq \mathbb{E}\left[\delta_t^2 - \delta_{t+1}^2\right] + 2\beta^2 \sigma^2 + \mathbb{E}\left[\frac{\beta^{4a-3} \|\mathbf{z}_t\|^2}{14G^{1/2}}\right]. \tag{37}$$

Taking summation on both sides of (37) over $t = 1, \ldots, T$, we have

$$\sum_{t=1}^{T} \mathbb{E}\left[\delta_t^2\right] \leq \frac{\mathbb{E}\left[\delta_1^2\right]}{\beta} + 2\beta\sigma^2 T + \sum_{t=1}^{T} \mathbb{E}\left[\frac{\beta^{4a-4} \|\mathbf{z}_t\|^2}{14}\right]$$

$$= \frac{\mathbb{E}\left[\delta_1^2\right]}{\beta} + 2\beta\sigma^2 T + \sum_{t=1}^{T} \mathbb{E}\left[\frac{\beta^a \|\mathbf{z}_t\|^2}{14G^{1/2}}\right] \tag{38}$$

$$\leq \frac{\mathbb{E}\left[\delta_1^2\right]}{\beta} + 2\beta\sigma^2 T + \sum_{t=1}^{T} \mathbb{E}\left[\frac{\beta^a \|\mathbf{z}_t\|^{3/2}}{14}\right],$$

where the equality holds since $a = 4/3$ and last inequality holds since $\|\mathbf{z}_t\| \leq G$.

Taking expectation on both sides of (36) and combining (38), we have

$$\sum_{t=1}^{T} \mathbb{E}\left[\|\mathbf{z}_t\|^{3/2} + \|\nabla F(\mathbf{w}_t)\|^{3/2}\right]$$

$$\leq \frac{12L(F(\mathbf{w}_1) - F_*)}{\beta^a} + \frac{14\mathbb{E}\left[\delta_1^2\right]}{\beta^{1+a}} + \frac{28\beta\sigma^2 T}{\beta^a} + \sum_{t=1}^{T} \mathbb{E}\left[\|\mathbf{z}_t\|^{3/2}\right] + 4\beta^{3a}T.$$

As a result, we have

$$\frac{1}{T}\sum_{t=1}^{T} \mathbb{E}\left[\|\nabla F(\mathbf{w}_t)\|^{3/2}\right] \leq \frac{12L(F(\mathbf{w}_1) - F_*)}{\beta^a T} + \frac{14\mathbb{E}\left[\delta_1^2\right]}{\beta^{1+a}T} + \frac{28\beta\sigma^2}{\beta^a} + 4\beta^{3a}. \qquad \square$$

## F    RELATED WORK

**Adaptive Gradient Methods**    Adaptive gradient methods were first proposed in the framework of online convex optimization (Duchi et al., 2011; McMahan & Streeter, 2010), which dynamically incorporate knowledge of the geometry of the data to perform more informative gradient-based learning. This type of algorithm was proved to have fast convergence if stochastic gradients are sparse (Duchi et al., 2011). Based on this idea, several other adaptive algorithms were proposed to train deep neural networks, including Adam (Kingma & Ba, 2014), Amsgrad (Reddi et al., 2019), RMSprop (Tieleman & Hinton, 2012). There are many work trying to analyze variants of adaptive gradient methods in both convex and nonconvex case (Chen et al., 2018a; 2019; 2018b; Luo et al., 2019; Chen et al., 2018a;b; Ward et al., 2019; Li & Orabona, 2019; Chen et al., 2019). Notably, all of these works are able to establish faster convergence rate than SGD, based on the assumption that stochastic gradients are sparse. However, this assumption may not hold in deep learning. In contrast, our algorithm can have faster convergence than SGD even if stochastic gradients are not sparse, since our algorithm's new data-dependent adaptive complexity does not rely on the sparsity of stochastic gradients.

**Variance Reduction Methods**    Variance reduction is a technique to achieve fast rates for finite sum and stochastic optimization problems. It was first proposed for finite-sum convex optimization (Johnson & Zhang, 2013) and then it was extended in finite-sum nonconvex (Allen-Zhu & Hazan, 2016; Reddi et al., 2016; Zhou et al., 2018) and stochastic nonconvex (Lei et al., 2017; Fang et al., 2018; Wang et al., 2019; Pham et al., 2020; Cutkosky & Orabona, 2019) optimization. To prove faster convergence rate than SGD, all these works make the assumption that the objective function is an average of individual functions and each one of them is smooth. In contrast, our analysis does not require such an assumption and to achieve a faster-than-SGD rate.

**Other Related Work**    Arjevani et al. (2019) show that SGD is optimal for stochastic nonconvex smooth optimization, if one does not assume that every component function is smooth. There are recent work trying to establish faster rate than SGD, when the Hessian of the objective function is Lipschitz (Fang et al., 2019; Cutkosky & Mehta, 2020). There are several empirical papers, including LARS (You et al., 2017) and LAMB (You et al., 2019)), which utilize both moving average and normalization for training of deep neural networks with large-batch sizes. Zhang et al. (2020) consider an algorithm for finding stationary point for nonconvex nonsmooth problems. Levy (2017) considers convex optimization setting and design algorithms which adapts to the smoothness parameter. Liu et al. (2019) introduced Rectified Adam to alleviate large variance at the early stage. However, none of them establish data-dependent adaptive complexity as in our paper.

## G    ADAM$^+$ WITH FIXED LEARNING RATE

We report the results on image classification with CIFAR10 on ResNet18. We further considered the Adam$^+$ with fixed learning rate 0.1 and do not employ any learning rate annealing scheme. We report our result on Figure 6, in which "Adam+ Fixed Stepsize" uses the default setting of Algorithm 1. As we can see from the Figure, Adam$^+$ with fixed stepsize still outperforms Adam,

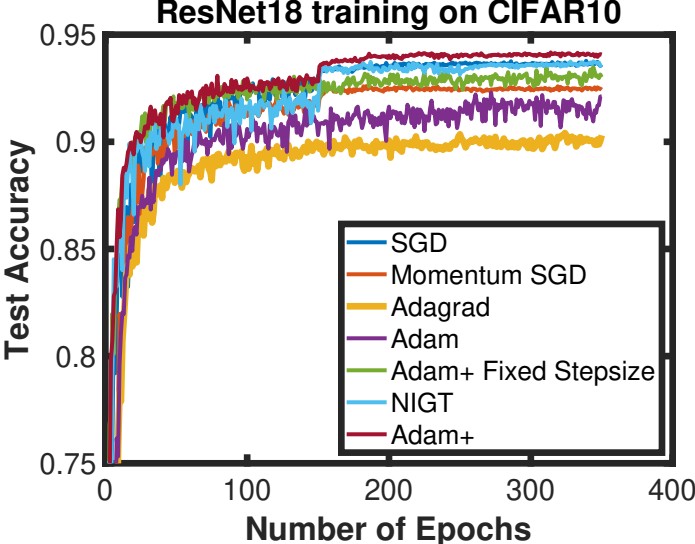

Figure 6: Comparison of Adam$^+$ and Adam$^+$ with Fixed Stepsize

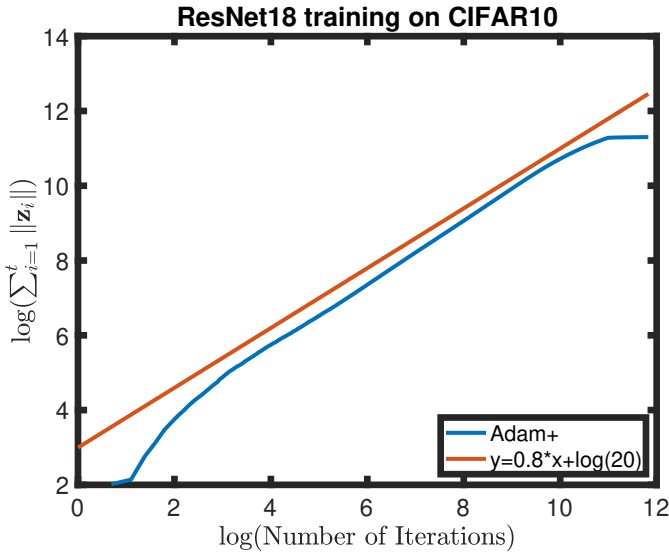

Figure 7: $\log(\sum_{i=1}^{t} \|\mathbf{z}_i\|)$ versus $\log(t)$

Adagrad and momentum SGD. If we use the same learning rate scheduler of SGD for Adam$^+$, then Adam$^+$ performs the best.

# H   GROWTH RATE ANALYSIS OF $\sum_{i=1}^{t} \|\mathbf{z}_i\|$

We provide the log-log plot $(\log(\sum_{i=1}^{t} \|\mathbf{z}_i\|)$ versus $\log(t))$ for the ResNet18 training on CIFAR10 experiment, as illustrated in Figure 7. We can see that the slope is around $0.8$. Although the slope is not small, we can see from Figure 5 that the slope becomes almost zero after the iteration $6 \times 10^4$ (which corresponds to the epoch number 154). At this particular epoch, the training and test accuracy are not the best so we need to keep the training until epoch 350. Then our algorithm Adam$^+$ is able

to take advantage of the slow growth rate of $\sum_{i=1}^{t} \|\mathbf{z}_i\|$ for large $t$ and enjoys faster convergence, which is consistent with Figure 1.

