# OpenReview forum: "Adam$^+$: A Stochastic Method with Adaptive Variance Reduction"
_ICLR.cc/2021/Conference — Reject_

### Official Review · AnonReviewer1 · 2020-10-28
**Adaptive aspects of the algorithm and improvement over SGD are not clear**

**Rating:** 4
**Confidence:** 4

**Review:**

This paper proposes a new optimizer called Adam+, with two main distinctions from standard Adam template: 1) the first order moment estimate is computed using the gradient evaluated at an extrapolated iterate. 2) the step size is scaled with the square root of the norm of the first order moment, rather than the exponential moving average (EMA) in the previous work. Under Lipschitz continuous gradient, Hessian and bounded gradient assumptions, the complexity for finding a point with small gradient norm is $\epsilon^{-3.5}$, which is a SOTA complexity. The practical performance of the algorithm is evaluated in a range of different tasks and the performance is shown to be consistently promising and comparable to SGD/Momentum SGD.

Strengths:
- I think the algorithm template, and the connection to variance reduction is interesting. In particular, I saw a similar algorithm in [1, Alg. 2]. Of course the algorithm in [1] is designed for a more general setting, so it has important differences. But in my opinion, the fact that a similar algorithm works in such a general setting as [1], shows the strength and potential of Adam+.

- The authors consider a large variety of tasks in different domains (Table 2) which is important to judge the consistency of the method in practice.

- The paper is clear and well-written, the comparison with previous work is fair and comprehensive, to my knowledge.

Weaknesses and suggestions for improvement: I have several concerns about the potential impact of both theoretical and practical results. Mainly:

- By referring to Wilson et al., 2017, the authors argue that diagonal step sizes in adaptive algorithms hurt generalization. First, I find this claim rather vague, as there has been many followups to Wilson et al., 2017, so I suggest the authors to be more precise and include more recent observations. Moreover, one can use non-diagonal versions of these algorithms. For example, see [2 and Adagrad-norm from Ward et al., 2019], it is easy to consider similar non-diagonal versions of Adam/AMSGrad/Adagrad with first order momentum (a.k.a. AdamNC or AdaFOM), then, are these algorithms also supposed to have good generalization? I think it is important to see how these non-diagonal adaptive methods behave in practice compared to SGD/Adam+ for generalization to support the authors' claim.

- I think the algorithm seems more like an extension of momentum SGD, than Adam.

- It is nice to improve \eps^{-4} complexity with Lipschitz Hessian assumption, but what happens when this assumption fails? Does Adam+ get standard \epsilon^{-4}?

- From what I understand in remark after Lemma 1, the variance reduction is ensured by taking $\beta$ to $0$. The authors use $1/T^{a}$ for some $a\in(0,1)$. Here, I have several questions. First, how does such a small $\beta$ work in practice? If in practice, a larger $\beta$ works well and theory requires $\beta\to 0$ for working, it shows to me that theoretical analysis of the paper does not translate to the practical performance. When one uses $\beta$ values that work well in practice, does the theory show convergence?

- Related to the previous part, I am also not sure about "adaptivity" of the method. The authors need to use Lipschitz constants $L, L_H$ to set step sizes. Moreover $\beta$ is also fixed in advance, depending on horizon $T$, which is the main reason to have variance reduction on $\|z_t-\nabla f(w_k)\|$. So, I do not understand what is adaptive in the step size or in the variance reduction mechanism of the method.

- For experiments, the authors say that Adam+ is comparable with "tuned" SGD. However, from the explanations in the experimental part, I understand that Adam+ is also tuned similar to SGD. Then, what is the advantage compared to SGD? If one needs the same amount of tuning for Adam+, and the performance is similar, I do not see much advantage compared to SGD. On this front, I
suggest the authors to show what happens when the step size parameter is varied, is Adam+ more robust to non-tuned step sizes compared to SGD?

To sum up, I vote for rejection since 1) the analysis and parameters require strict condition, 2) it is not clear if the analysis illustrates the practical performance (very small $\beta$ is needed in theory), 3) practical merit is unclear since the algorithm needs to be tunes similar to SGD and the results are also similar to SGD.

[1] Zhang, Lin, Jegelka, Jadbabaie, Sra, Complexity of Finding Stationary Points of Nonsmooth Nonconvex Functions, ICML 2020.
[2] Levy, Online to Offline Conversions, Universality and Adaptive Minibatch Sizes, NIPS 2017.

======== after discussion phase ==========

I still think that the merit of the method is unclear due to reasons: 1) It is not clear how the method behaves without Lipschitz Hessian assumption. 2) The method only obtains the state-of-the-art complexity of $\epsilon^{-3.5}$ with large mini-batch sizes and the complexity with small mini-batch sizes (section 2.1) is suboptimal (in fact drawbacks such as this needs to be presented explicitly, right now I do not see enough discussions about this.). 3) Adaptive variance reduction property claimed by the authors boils down to picking "small enough" $\beta$ parameter, which in my opinion takes away the adaptivity claim and is for example not the case in adaptive methods such as AdaGrad. 4) The comparison with AdaGrad and Adam with scalar step sizes is not included (the authors promised to include it later, but I cannot make a decision about these without seeing results) and I am not sure if Adam+ will bring benefits over them. 5) Presentation of the paper needs major improvements. I recommend making the remarks after Lemma 1 and theorems clearer, by writing down exact expressions and the implications of these (for example remarks such as "As the algorithm converges with $\mathbb{E}[\|\nabla F(w_)\|^2]$ and $\beta$ decreases to zero, the variance of $z_t$ will also decrease" can be made more rigorous and clearer, by writing down exactly the bound for the variance of $z_t$ by iterating the recursion written with $\mathbb{E}\delta_{t+1}$ and highlighting what each term does in the bound. This way will be much easier for readers to understand your paper).

Therefore, I am keeping my score.

---

> ### Author Response · Authors · 2020-11-16
> **Thank you for your constructive feedback. We have included new results in the updated version of this paper.**
>
> Thanks for your insightful comments. Please check our responses below.
>
> Q1: It is important to see how these non-diagonal adaptive methods behave in practice compared to SGD/Adam+ for generalization to support the authors’ claim.
>
> A: Thank you for mentioning these papers. We have cited all of them in the updated version. We expect to include the comparison with these algorithms in the final version and make this point more clear.
>
>
> Q2: How does a small $\beta$ work in practice?
>
> A: Indeed small $\beta$ works in practice. Our $\beta$ roughly corresponds to $1-\beta$ in momentum SGD. Usually in momentum SGD, $1-\beta$ is set to be 0.9 (for example, in the resnet paper (He et al. 2016)), and hence it corresponds to $\beta=0.1$ in our case.
>
> Q3: I do not understand what is adaptive in the stepsize or in the variance reduction mechanism of the method?
>
> A: This paper is not adaptive in the stepsize as in (Ward et al. 2019) but shares the similar spirit of the original Adagrad paper (Duchi et al. 2011). In (Ward et al. 2019), Adagrad-Norm does not require any knowledge of smoothness constant but it also does not prove that the algorithm enjoys faster convergence by taking advantage of the geometry in the data (e.g., the presence of sparsity in stochastic gradients) as in (Duchi et al. 2011). For our Adam+, we show that it is adaptive to the geometry of the data which is $\sum_{t=1}^T \left\|z_t\right\|$ as shown in Theorem 1 and Theorem 2. In Theorem 2, we show that the noise in the gradient estimator $z_t$ is vanishing in an ergodic sense, so this means that the Adam+’s convergence benefits from the variance reduction property of the gradient estimator of $z_t$. If $\sum_{t=1}^T \left\|z_t\right\|$ grows in a slow rate in practice (which we empirically verified in Figure 5), then Adam+ benefits from it and enjoys faster convergence. Please check the Remarks under Theorem 2.
>
>
> Q4: Adam+ is also tuned similar to SGD.
>
> A: Adam+ is *NOT* tuned similar to SGD, since we never tune the learning rate schedule of Adam+. We just simply choose the same learning rate schedule of Adam+ to be the same as in best-tuned SGD. If we fix the initial learning rate in Adam+ and do not employ any learning rate annealing (which is the same as the default setting described in Algorithm 1), the performance is still better than Adam and Adagrad. We include such an experiment in the Appendix G to illustrate this point.

---

### Official Review · AnonReviewer4 · 2020-10-28
**Variant of Adam with fractional self-normalization**

**Rating:** 5
**Confidence:** 3

**Review:**

Summary: This paper proposes the Adam+ algorithm that maintains an exponential moving average of the first moment and normalizes it by its $p$-th moment for some $p \in (1/2, 1)$. When $p = 2/3$, with appropriate hyperparameters, Adam+ achieves the state-of-the-art complexity $O(1/\epsilon^{3.5})$ to obtain an approximate first-order stationary point for smooth objectives with smooth Hessians. The proof technique is similar to that for SCGD by Wang et al. (2017), which construct a Lyapunov function for algorithms of this kind.

Pros:
(1) The idea of using fractional normalization is interesting.

(2) The proof involves bounding the (4/3)-th moment of the average gradient norm, which is novel to me.

(3) The algorithm performs well on the examples considered in the paper.

Concerns:
(1) The rate $O(1/\epsilon^{3.5})$ requires the inner batch size to be $O(\epsilon^{-1.5})$. The requirement of large batch sizes seem to be essential for the proof. But it is not a very realistic setting. By contrast, the NIGT algorithm (Cutkosky and Mehta, 2020) does not require large batches while achieves the same convergence rate. Moreover, for both Theorem 1 and 3, an extra assumption that $||\nabla F(x)||\le G$ is imposed, which is not required by NIGT. Is the analysis for Adam+ unnecessarily loose, given that Adam+ and NIGT are almost identical?

(2) Adam+ seems to be not scale-invariant even if the term $\epsilon_0$ is ignored in $\eta_t$. When $f$ becomes $Cf$, $L$ becomes $CL$, and $\nabla g$ becomes $C\nabla g$. Since the scale of $w$ does not change, we should expect the sequence $w_{k}$ to be invariant to $C$. Equivalently, we want $\eta_t z_t$ to be invariant. If Adam+ with $p = 2/3$ is scale-invariant, then $z_t$ becomes $C z_t$, in which case $\eta_t z_t \propto \alpha C^{1/3}z_{t} / ||z_{t}||^{2/3}$. As a result, $\alpha$ should scale as $C^{-1/3}$. However, in Theorem 3, $(CL)^2 \alpha^4 = O(1)$, implying that $\alpha$ scales as $C^{-1/2}$. This contradicts with the scale-invariance. Therefore, Adam+ is not scale-invariant. If my argument is correct, Adam+ seems to be problematic. Could you clarify how scale invariance can be achieved? Does $\beta$ or other parameters implicitly depend on the smoothness parameter?

(3) The $O(poly(1/\epsilon))$ in Theorem 2 is $O(\epsilon^{-4.5})$, right? If so, please clarify this rate since $O(poly(1/\epsilon))$ is too vague.

(4) The sharp slope change in Figure 5 is unexpected. Is this due to the decreased step size? If the curve before the turning point is fitted by $T\rightarrow T^{\alpha}$, what is estimate of $\alpha$? I suspect that the estimated alpha is close to $1$. If so, the point made in Theorem 1 would be undermined.

(5) For experiments on CIFAR10 and CIFAR100, the performance of Adam seems to be very poor. For instance, NIGT substantially outperforms Adam, but even in Cutkosky and Mehta (2020) the gap is not as large as the presented one. Is the best tuned result presented for Adam?

Typos:

(1) In page 12 and 15, $\zeta_{t}^{(k)}$ should be $\zeta_{k}^{(t)}$,
$w_{t} = \sum_{k=0}^{t}\zeta_{k}^{(t)}\hat{w}_{t+1}$ should be $w_{t} = \sum_{k=0}^{t}\zeta_{k}^{(t)}\hat{w}_{k+1}$, and $z_{t+1} = \sum_{k=0}^{t}\zeta_{k}^{(t)}\nabla f(\hat{w}_{t+1}; \xi_{t+1})$ should be $z_{t+1} = \sum_{k=0}^{t}\zeta_{k}^{(t)}\nabla f(\hat{w}_{k+1}; \xi_{k+1})$ (The equations fail to be displayed for no reason)

(2) Page 12: $||z_{t} - \nabla F(w_{t})||^2\le (Lm_{t} + ||n_{t}||)^2$ should be $||z_{t} - \nabla F(w_{t})||^2\le (L_{H}m_{t} + ||n_{t}||)^2$.

(3) Page 12-13, equation (5)-(6): $\sigma^2$ should be $\sigma_{m}^2$.

(4) Page 13, the last equation: the first term should has a constant $8$.

(5) Page 15, equation (15): $n_{k+1}$ and $n_{k}$ should be $n_{t+1}$ and $n_{t}$, respectively.

---

> ### Author Response · Authors · 2020-11-16
> **Thank you for your insightful comments. We have updated our paper according to your suggestions.**
>
> Thanks for your careful review for our papers. We will address your concerns below. All typos are fixed in the revised version.
>
>
> Q1: The requirement of large minibatch is not a very realistic setting.
>
> A: We agree. However, we want to highlight that our main contribution is section 2.1, which establishes adaptive convergence results and does not require large minibatch. In our experiment, we also implemented the algorithm in section 2.1. The purpose of section 2.2 is to introduce a general variant of our algorithm and show that it is able to match the complexity of NIGT in the worst case.
>
>
>
>
> Q2:  For both Theorem 1 and Theorem 3, an extra assumption assumption that $\|\nabla F(x)\|\leq G$ is imposed which is not required by NIGT.
>
>
> A: The proof of Theorem 3 does not need the bounded gradient assumption, please refer to the Appendix D for the proof. The proof of Theorem 1 only utilizes the bounded gradient assumption in the very last step for proving (inequality (10), page 13, Appendix B), which is to make the bound easier to understand. Without the bounded gradient assumption, we also have the adaptive convergence result, which is presented in the (inequality (9), page 13, Appendix B). Please check Appendix B for the proof. We will make it more clear in the final version.
>
>
>
> Q3: Adam+  with $p=2/3$ seems to be not scale-invariant.
>
> A:  If $\beta$ is not scale-invariant, then Adam+ with $p=2/3$ is indeed not scale-invariant. However, if we allow $\beta$ to depend on $L$, then it is easy to get a scale-invariant algorithm. For example, if we choose $\beta^{3/4}=O(L^{1/6}\epsilon^{1/2})$, then the algorithm is scale-invariant with $O(\epsilon^{-3.5})$ complexity (please check the inequality (30) at Page 18 in Appendix D). Moreover, our Theorem 2 provides a scale-invariant guarantee for Adam$^+$ with $p=1/2$. Since $\alpha\leq 1/\sqrt{L_H}$, so it scales as $C^{-1/2}$, and hence $\eta_t$  in Algorithm 1 scales as $C^{-1}$, so $\eta_t z_t$ is scale-invariant. In our experiment, we also use Algorithm 1 instead of the variant with $p=2/3$.
>
>
> Q4: $O(poly(1/\epsilon))$ is too vague in Theorem 2.
>
> A: Thanks for the comments. The $O(\epsilon^{-4.5})$ result holds under the case that $\beta=T^{-2/3}$. We have changed the statement of Theorem 2 in the updated version.
>
>
> Q5: The sharp slope change in Figure 5 is unexpected. If the curve before the turning point is fitted by $T^\alpha$, what is the value of $\alpha$?
>
>
> A: The value of $\alpha$ is roughly $0.8$ in this experiment before the turning point. The growth rate of the quantity should not scale linearly in $T$ in theory as well, since otherwise the algorithm would not converge and it would be contradictory to Theorem 2. More importantly, we also want to highlight that our algorithm Adam$^+$ is able to take advantage of the slow growth rate of $\sum_{i=1}^{t}\|z_i\|$ for large $t$, since small $t$ at the turning point does not provide good classification performance so we need to continue the training process. We have included more details in Appendix H.
>
>
> Q6: The performance of Adam in CIFAR10 and CIFAR100, the results are very poor. NIGT substantially outperforms Adam but the gap is not as large as the presented one. Is it the best tuned results for Adam?
>
> A:  These results are the best tuned results for Adam. We tuned initial learning rates of Adam from $\{0.1, 0.01, 0.001\}$ in our experiment and the best one is $0.001$. We have reported the best parameter setting for every experiment in the updated version, please check our section 3. NIGT paper does not consider the same experimental settings as in our paper so it is hard to say why the gap is large in our experiments. We conjecture that it is due to the different learning rate scheduler. For learning rate, they use polynomial decay (i.e., $\eta_t=\eta_0 (1-t/T)$) while we use stagewise exponential decay. The latter one is used in the original AlexNet and ResNet paper, and there are also some theoretical papers justifying why the latter one is better ([ref1, ref2]).
>
> [ref1] Ge et al. The Step Decay Schedule: A Near Optimal, Geometrically Decaying Learning Rate Procedure For Least Squares. NeurIPS 2019.
>
> [ref2] Yuan et al. Stagewise training accelerates convergence of testing error over SGD. NeurIPS 2019.

---

### Official Review · AnonReviewer3 · 2020-10-29
**An Adam variant by replacing 2nd order momentum with 1st order momentum**

**Rating:** 6
**Confidence:** 3

**Review:**

This paper propose a new variant of ADAM, which replace 2nd order momentum with 1st order momentum.

Pros:
1. It saves the number of saved parameters (i.e., without using 2nd order momentum).
2. This paper provide convergence analysis. The author is able to derive a data dependent convergence rate under mild conditions.
3. The theoretical analysis and implication is very clear. The variance reduction property of the 1st order momentum provides a good characterization of the convergence of the algorithm.
4. Provide comprehensive empirical study on various benchmark datasets and neural networks in computer vision and natural language processing. The proposed ADAM+ converge faster compared with competitors.

Cons/Questions:
1. Section 2.2, what does it mean by "Large Mini-Batch"
2. In addition to the training curves, it is better to present a quantitative empirical results (for specific metrics) when comparing different optimizers. E.g., final classification accuracy on CIFAR-10
3. In the experiments of WikiText2, the convergence of ADAM is much faster than the competitor? What are the learning rate used for ADAM and other competitors? It would be better to report the best hyper-parameter combination for each experiment.
4. A hyper-parameter sensitivity analysis of ADAM+ is needed.

Minor:
1. missing reference: On the Variance of the Adaptive Learning Rate and Beyond

---

> ### Author Response · Authors · 2020-11-16
> **Thank you for your valuable comments. We have updated the manuscript accordingly.**
>
> Thanks for your constructive feedback. Please check our responses below.
>
> Q1: In section 2.2, what does it mean by “large minibatch”?
>
> A: “large minibatch” means that the minibatch size $m$ could depend on $\epsilon$. For example, in Theorem 3, $m=O\left(\epsilon^{-3/2}\right)$. Note that the large minibatch is only needed in section 2.2 but is not required in section 2.1.
>
>
> Q2: It is better to present quantitative empirical results when comparing different optimizers. E.g. final classification accuracy on CIFAR-10.
>
> A: Thanks for the suggestion. We will include the final results in a Table in the final version.
>
>
> Q3: It would be better to report the best hyper-parameter combination in each experiment.
>
> A: We have included all hyperparameter combinations for each experiment in section 3 (marked in red).
>
>
> Q4: A hyperparameter sensitivity analysis of Adam+ is needed.
>
> A: Thanks for the suggestion. According to our observation, Adam+ is not sensitive in choosing parameters. In our Algorithm 1, we provide good default settings for tested machine learning problems: $\alpha=0.1, a=1, \beta=0.1, \epsilon=10^{-8}$. We will compare different parameters and results in the final version.

---

### Official Review · AnonReviewer2 · 2020-10-29
**A new optimizer for deep learning**

**Rating:** 5
**Confidence:** 4

**Review:**

Objective of the paper: The paper proposes a new optimizer "Adam+" that computes the first moment estimate at extrapolated points and the step size is normalized by the root of the norm of the first moment estimate. The paper establishes a convergence theory for Adam+ and conducts experiments on different deep learning tasks to demonstrate the advantage of Adam+.

Strong points. The paper proposes a novel algorithm to train the deep neural network. It has both theoretical guarantee and empirical evidence to justify the advantage of Adam+. The paper logic is clear  and well-organized.

Weak points:
1. The theoretical guarantee is problematic.  In Theorem 1, the right hand side of (1) has $\sum_t \|z_t\|$ which could scale linearly with $T$. The right hand side and the left hand side share the same structure sum of gradients and hence it is not appropriate to claim this bound meaningful in terms of $T$. This is different from the Adam AdaGrad's proof, where $\|g_{1:T,i}\|$ scales with $\sqrt{T}$ and hence produces a $1/\sqrt{T}$ regret.
2.  The empirical result does not cover Transformer like models, which achieve SOTA performance on language understanding and favor Adam optimizer. Moreover, it is known that the performance of Adam is sensitive to the learning rate [1].   For fair comparison, the paper should also tune the hyper-parameters of Adam at least its learning rate.


I do not recommend the publication for now.

Minor,
In Abstract: at extrapolated data points. --> at extrapolated points

[1]  Choi, Dami, et al. "On empirical comparisons of optimizers for deep learning." arXiv preprint arXiv:1910.05446 (2019).

After rebuttal

Thanks for the feedback. However, I am not persuaded by the answers of Q1 and Q2 and would keep the score unchanged.

---

> ### Author Response · Authors · 2020-11-16
> **Thank you for your review.**
>
> Thanks for your valuable comments. Please check our answers below.
>
> Q1: The right hand side of of (1) has $\sum_t |z_t|$ which could scale linear in $T$.
>
> A: Thanks for this insightful question. Theorem 1 itself indeed seems not to be able to guarantee the convergence in the worst case. However, we have proved Theorem 2, which indicates that our algorithm always converges (the ergodic convergence of $\left\|\nabla F(\mathbf{w}_t)\right\|^{3/2}$), and the variance gets smaller when $T$ gets larger (the ergodic convergence of $ \delta_t^{3/2}$)). So it rules out the possibility of the linear growth of $\sum_t |z_t|$, since otherwise the algorithm would not converge. Note that $E\left\[\frac{1}{T}\sum_{t=1}^{T}\left\|z_t\right\|\right]\leq E\left[\sum_{t=1}^{T}\left(\delta_t+\left\|\nabla F(\mathbf{w}_t)\right\|\right)\right]$, and $\delta_t$ is proved to have ergodic convergence to any small constant $\epsilon$, so Adam+’s convergence benefits from the variance reduction property of the gradient estimator $\mathbf{z}_t$ and $\sum_t |z_t|$ must grow sub-linearly in $T$ even in the worst case . Please check the Remarks below the Theorem 2. In addition, we have also did empirical studies in Figure 5 (page 8) to show that the growth rate of $\sum_t \|\mathbf{z}_t\|$ is indeed slow in practice.
>
>
>
>
>
> Q2: The empirical result does not cover Transformer like models, which achieve SOTA performance on language understanding and favor Adam optimizer.
>
> A: In the transformer paper ([ref1]), their Adam optimizer (section 5.3, formula (3)) uses a specifically designed learning rate, i.e.,
> $lrate=d_{\text{model}}^{-0.5}\cdot\min\left(stepnum^{-0.5}, stepnum\cdot warmupsteps^{-1.5}\right)$.
> Note that this learning rate depends on the warmup steps, number of steps, with different exponents. This is very different from the original Adam optimizer. Our main focus of this paper is comparing our algorithm with the original Adam optimizer on various domains (CV, NLP, ASR).
>
> [ref1] Vaswani et al. Attention is all you need. NIPS 2017.
>
>
>
> Q3: The paper should also tune the hyperparameters of Adam at least its learning rate.
>
> A: We indeed tuned the learning rate of Adam for every experiment. We tuned the initial learning rate of Adam from $(0.1, 0.01, 0.001)$ in section 3.1, $(1.0, 0.1, 0.01, 0.001)$ in section 3.2, and $(0.1, 0.01, 0.001)$ in section 3.3, as we mentioned in the paper. We also reported the best parameter setting for every algorithm in the updated version.

---

### Author Response · Authors · 2020-11-16
**General Comments**

Dear reviewers,

Thanks for all the comments. We have updated our manuscript per reviewers' suggestions. All updates are marked in red. The main summary of our updates are:

1. We report all the best selected hyperparameters for all algorithms in section 3, as suggested by R2, R3.

2. We consider Adam$^+$ with a fixed learning rate and add a Figure 6 in in Appendix G. We show that Adam$^+$ with fixed learning rate still outperforms Adam and Adagrad in ResNet18 training on CIFAR10, per R1’s suggestion.

3. We provide the growth rate analysis of $\sum_{t=1}^{T}\|z_i\|$ in Appendix H, per R4’s suggestion.

4. We have changed the statement of Theorem 2 and also explained the scale-invariant property of our algorithm, as suggested by R4.

5. We have added all missing references and addressed all minor issues raised by reviewers.

---

### Decision · Program_Chairs · 2021-01-07
**Final Decision**

**Decision:**

Reject

**Comment:**

This paper proposes a new variant of adaptive stochastic gradient method that has notable differences from Adam, and claims the advantage of adaptive variance reduction. While the algorithm construction looks novel, there are several concerns by the expert reviewers on the theoretical results of the paper, including lack of clarity and its guidance/relevance to the practical performance. There are also questions on the practical merit of the paper pointing to in limitations on the numerical experiments. I recommend rejection of the paper in the current form, and hope the reviews can help the authors to improve it in both theoretical and empirical aspects.